EPN-Repro2: A reference GNSS tropospheric dataset over Europe.

Rosa Pacione [(1)], Andrzej Araszkiewicz [(2)], Elmar Brockmann [(3)], Jan Dousa [(4)]

[(1)]   e-GEOS S.p.A, ASI/CGS, Italy
[(2)]   Military University of Technology, Poland
[(3)]   Swiss Federal Office of topography swisstopo
[(4)]   New Technologies for the Information Society, Geodetic Observatory Pecný, RIGTC, Czech
Republic

*Correspondence to*: Rosa Pacione (rosa.pacione@e-geos.it)

**Abstract.** The present availability of 18+ years of GNSS data belonging to the EUREF Permanent

Network (EPN, http://www.epncb.oma.be/) is a valuable database for the development of a climate

data record of GNSS tropospheric products over Europe. This data record can be used as a reference

for a variety of scientific applications (e.g. validation of regional Numerical Weather Prediction

reanalyses and climate model simulations) and has a high potential for monitoring trends and the

variability in atmospheric water vapour. In the framework of the EPN-Repro2, the second

reprocessing campaign of the EPN, five Analysis Centres homogenously reprocessed the EPN

network for the period 1996-2014. A huge effort has been made for providing solutions that are the

basis for deriving new coordinates, velocities and tropospheric parameters for the entire EPN. The

individual contributions are then combined to provide the official EPN reprocessed products. This

paper is focused on the EPN-Repro2 tropospheric product. The combined product is described

along with its evaluation against radiosonde data and European Centre for Medium-Range Weather

Forecasts (ECMWF) reanalysis (ERA-Interim) data.

**1. Introduction**

The EUREF Permanent Network (Bruyninx et al., 2012; Ihde et al., 2013) is the key geodetic

infrastructure over Europe, currently made up by over 280 continuously operating GNSS [Global

Navigation Satellite Systems as USA's NAVSTAR Global Positioning System (GPS) and Russia's

Global'naya Navigatsionnaya Sputnikovaya Sistema (GLONASS)] reference stations, and

maintained on a voluntary basis by EUREF (International Association of Geodesy Reference Frame

Sub-Commission for Europe, http://www.euref.eu) members. Since 1996, GNSS data collected at

the EUREF Permanent Network have been routinely analysed by several (currently 16) EPN

Analysis Centres (Bruyninx et al., 2015). For each EPN station, observation data along with

metadata information as well as precise coordinates and tropospheric Zenith Total Delay (ZTD)

parameters are publicly available. Since June 2001, the EPN Analysis Centres (AC) routinely

estimate ZTD in addition to station coordinates. The ZTD, available in daily SINEX TRO files, are

used by the coordinator of the EPN tropospheric product to generate each week the final EPN

solution containing the combined tropospheric estimates with an hourly sampling rate. The

coordinates, as a necessary part of this file, are taken from the EPN weekly combined SINEX file (http://www.iers.org/IERS/EN/Organization/AnalysisCoordinator/SinexFormat/sinex.html). Hence, stations without estimated coordinates in the weekly SINEX file are not included in the combined troposphere solution. The generation of the weekly combined products is done for the routine analysis. Plots of the ZTD time series and ZTD monthly means as well as comparisons with respect to radiosonde data are available in a dedicated section at the EPN Central Bureau web site (http://www.epncb.oma.be/_productsservices/sitezenithpathdelays/). Radiosonde profiles are provided by EUMETNET (European Meteorological Services Network) as an independent dataset to validate GNSS ZTD data, and are exchanged between EUREF and EUMETNET for scientific purposes, based on a Memorandum of Understanding between the two mentioned organisations (http://www.euref.eu/documentation/MoU/EUREF-EUMETNET-MoU.pdf).

However, such time series are affected by inconsistencies due to updates of the reference frame and the applied models, implementation of different mapping functions, use of different elevation cut-off angles and any other updates in the processing strategies that causes inhomogeneities over time. To reduce processing-related inconsistencies, a homogenous reprocessing of the whole GNSS data set is mandatory and, for doing it properly, a well-documented, long-term metadata set is required.

This paper focuses on the tropospheric products obtained in the framework of the second EPN Reprocessing campaign (hereafter EPN-Repro2), for which, using the latest available models and analysis strategy, GNSS data of the entire EPN network have been homogeneously reprocessed for the period 1996-2014. The EPN homogeneous long-term GNSS time series can be used as a reference dataset for a variety of scientific applications in meteorological and climate research. Ground-based GNSS meteorology (Bevis et al.,1992) is very well established in Europe and dates back to the 90s, starting with the EC 4th Framework Program (FP) projects WAVEFRONT (GPS Water Vapour Experiment For Regional Operational Network Trials) and MAGIC (Meteorological Applications of GPS Integrated Column Water Vapour Measurements in the western Mediterranean, Haase et al., 2001). Early in this century, the ability to estimate ZTDs in Near Real Time has been demonstrated (COST-716, 2005), and the EC 5th FP scientific project TOUGH (Targeting Optimal Use of GPS Humidity Measurements in Meteorology, 2003-2006) was funded. Since 2005, the operational production of tropospheric delays has been coordinated and monitored by the EUMETNET GNSS Water Vapour Programme (E-GVAP, 2005-2017, Phase I, II and III, http://egvap.dmi.dk). Guerova et al. (2016) report on the state-of-the-art and future prospects of the ground-based GNSS meteorology in Europe. On the other hand, the use of ground-based GNSS long-term data for climate research is still an emerging field.

Promoting the use of reprocessed long-term GNSS-based tropospheric delay data sets for climate research is one of the objectives of the Working Group 3 'GNSS for climate monitoring' of the EU COST Action ES 1206 'Advanced Global Navigation Satellite Systems tropospheric products for monitoring severe weather events and climate (GNSS4SWEC)', launched for the period of 2013–2017. The Working Group 3 enforces the cooperation between geodesists and climatologists in order to generate recommendations on optimal GNSS reprocessing algorithms for climate applications, and to standardise for these applications the conversion method between propagation delay and atmospheric water vapour (Saastamoinen, 1973; Bevis et al., 1992; Bock et al. 2015). For climate applications, maintaining the long-term stability is a key issue. Steigenberger et al. (2007) found that the lack of consistencies over time due to changes in GNSS processing could cause inconsistencies of several millimetres in the GNSS-derived Integrated Water Vapour (IWV), making climate trend analysis very challenging. Jin et al. (2007) studied the seasonal variability of tropospheric GPS ZTD (1994-2006) over 150 international GPS stations and showed its relative trend in the northern and southern hemisphere as well as in coastal and inland areas. Wang and Zhang (2009) derived GPS Precipitable Water Vapour (PWV or PW) using the International GNSS Service (IGS, Dow et al.,2009) tropospheric products at about 400 global sites for the period 1997-2006 and analysed the PWV diurnal variations. Nilsson and Elgered (2008) reported on PWV changes from -0.2 mm to +1.0 mm in 10 years by using the data from 33 GPS stations located in Finland and Sweden. Sohn and Cho (2010) analysed the GPS Precipitable Water Vapour trend in South Korea for the period 2000-2009 and studied also the relationship between GPS PWV and temperature. A more thorough knowledge of atmospheric humidity, particularly in climate-sensitive regions, is essential to improve the diagnosis of global warming, and for the validation of climate predictions on which socio-economic response strategies are based. Suparta (2012) pointed out that the validation of PWV is an essential tool for solar-climate studies over a tropical region. Ning et al. (2013) used 14 years of GPS-derived IWV at 99 European sites to evaluate the regional Rossby Centre Atmospheric (RCA) climate model. GPS monthly mean data were compared against RCA simulations and ERA-Interim data. Averaged over the domain and the 14 years covered by the GPS data, they found IWV differences of about 0.47 $kg/m^2$ and 0.39 $kg/m^2$ for RCA-GPS and ERA-Interim-GPS, with standard deviations of 0.98 $kg/m^2$ and 0.35 $kg/m^2$, respectively. Alshawaf et al. (2017) found that GNSS IWV trends estimated at 113 GNSS sites in Europe, with 10 and 19 year temporal coverage, varies between -1.5 and 2 mm/decade with standard errors below 0.25 mm/decade. At these sites the ERA-Interim data analysed over 26 years show positive trends below 0.6 mm/decade, which correlate with the temperature trends.

Against this background, EPN-Repro2 is a unique dataset for the development of a climate data record of GNSS tropospheric products over Europe, suitable for analysing climate trends and variability, and calibrating/validating independent datasets at European and regional scales. However, although homogenously reprocessed, this time series still suffers from site-related inhomogeneities due, for example, to instrumental changes (receivers, cables, antennas, and radomes), changes in the station environment, etc. which might affect the analysis of the long-term variability (Vey et al., 2009). Therefore, to get realistic and reliable water vapour trend estimates such change points in the time series need to be detected and corrected for (Ning et al, 2016a).

This paper describes the EPN-Repro2 reprocessing campaign in Section 2. Section 3 is devoted to the combined solutions, i.e. the official EPN-Repro2 products, while in Section 4 the combined solution is evaluated w.r.t. radiosonde, ERA-Interim data and in terms of ZTD trends. The summary and recommendations for future reprocessing campaigns are drawn in Section 5.

## 2. EPN second reprocessing campaign

EPN-Repro2 is the second EPN reprocessing campaign organized in the framework of the special EUREF project "EPN reprocessing". The first reprocessing campaign, which covered the period 1996-2006 (Voelksen, 2011), involved the participation of all sixteen EPN Analysis Centres (ACs), reprocessing their own EPN sub-network. This strategy guaranteed that each site was processed by at least three ACs, which is an indispensable condition for providing a combined product. The second reprocessing campaign covered all the EPN stations, which were operated from January 1996 through December 2013. Then, the participating ACs decided to extend this period until the end of 2014 for tropospheric products. Data from about 280 stations in the EPN historical database have been considered. As of December 2014, 23% of EPN stations are between 15-18 years old, 26% are between 10-14 years old, 30% between 5-10 years old, and 21% less than 5 years old. Only five, over sixteen, EPN ACs (see Table 1) took part in EPN-Repro2, each providing at least one reprocessed solution. One of the goals of the second reprocessing campaign was to test the diversity of the processing methods in order to ensure the verification of the solutions. For this reason, the three main GNSS software packages Bernese (Dach et al., 2014), GAMIT (King et al., 2010) and GIPSY-OASIS II (Webb et al., 1997) have been used to reprocess the whole EPN network and, in addition, several variants have been provided. In total, eight individual contributing solutions, obtained using different software and settings, and covering different EPN networks, are available. Among them, three are obtained with different software and cover the full EPN network, while three are obtained using the same software (namely Bernese), but covering different EPN networks. In Table 2 the processing characteristics of each contributing solution are reported. Despite the

software used and the analysed networks, there are few diversities among the provided solutions, whose impact needs to be evaluated before performing the combination. In the reprocessing campaign all the ACs used for the GNSS orbits the CODE (Center for Orbit Determination in Europe) Repro2 product (Lutz et al., 2014), with one exception (see Table 2) where JPL (Jet Propulsion Laboratory) Repro2 products (Desai et al., 2014) are used. For tropospheric modelling two mapping functions are used: GMF-Global Mapping Function (Boehm et al., 2006a) and VMF1-Vienna Mapping Function (Boehm et al., 2006b), whose impact has been evaluated in Tesmer et al., 2007.

## 2.1    Impact of GLONASS data

During the reprocessing period, the Russian satellite system GLONASS became operational, and GLONASS observations are available since 2003. However, only from 2008 onwards the amount of GLONASS data (see Figure 1) is significant. The impact of GLONASS observations has been evaluated in terms of raw differences between ZTD estimates as well as on the estimated linear trend derived from the ZTD time series. As a matter of fact, GPS data (from the American navigation satellite system) are used by all ACs in this reprocessing campaign, while two of them (namely IGE and LPT) reprocessed GPS and GLONASS observations. Two solutions were prepared and compared, using the same software and the same processing characteristics, but different observation data: one with GPS and GLONASS, and one with GPS data only. The difference in ZTD trends (Figure 2) between a GPS-only and a GPS+GLONASS solution shows no significant rates for more than 100 stations (rates usually derived from more than 100000 ZTD differences). This indicates that the inclusion of additional GLONASS observations in the GNSS processing has a neutral impact on the ZTD trend analysis. Satellite constellations are continuously changing in time due to satellites being replaced and newly added for all systems. For instance, in the near future the inclusion of additional Galileo (navigation satellite system in Europe) and BeiDou (navigation satellite system in China) data will become operational in the GNSS data processing. These data will certainly improve the quality of the tropospheric products and our study here points out that the ZTD trends might be determined independently of the satellite systems used in the processing, and therefore might not introduce systematic changes in terms of ZTD trends.

## 2.2    Impact of IGS type mean and EPN individual antenna calibration models

According to the processing options listed in the EPN guidelines for the Analysis Centre (http://www.epncb.oma.be/_documentation/guidelines/guidelines_analysis_centres.pdf), EPN individual antenna calibration models have to be used instead of IGS type mean calibration models, when available. Currently, individual antenna calibration models are available at about 70 EPN

stations. As reported in Table 2, there are individual solutions carried out with IGS type mean antenna calibration models only (Schmid et al., 2015) while others use IGS type mean plus EPN individual antenna calibration models. Therefore, for the same station, there are contributing solutions obtained applying different antenna models. To evaluate the impact of using these different antenna calibration models on the ZTD, two solutions were prepared and compared, using the same software and the same processing, but different antenna calibration models: the first solution used the IGS type mean models only, and the second one used the individual calibrations whenever it was possible and the IGS type mean for the rest of the antennas. An example of the time series of the ZTD differences obtained between applying 'Individual' and 'Type Mean' antenna calibration models for the EPN station KLOP (Kloppenheim, Frankfurt, Germany) is shown in Figure 3. KLOP station is included in the EPN network since June, 2nd 2002, when a TRM29659.00 antenna from the Trimble Company with no radome was installed. In the forthcoming years, two major instrumentation changes occurred at the station: the first in June 27th 2007, when the previous antenna was replaced with a new type of Trimble antenna (TRM55971.00) and a dedicated hemisphere radome (TZGD) was installed, and a second change in June 28th 2013 with the installation of another type of Trimble antenna (TRM57971.00) and the same type of radome. For these three specific hardware sets the individual calibrations are available at the EPN Central Bureau (ftp://epncb.oma.be/pub/station/general/epnc_08.atx). Switching between phase centre corrections from type mean to individual (or vice versa) causes a disagreement in the estimated up component of the stations, as was mentioned by Araszkiewicz and Voelksen (2016), and as a consequence in their ZTD time series. Depending on the antenna model, the offset at station KLOP in the up component (vertical displacement) is $-5.2 \pm 0.5$ mm, $8.7 \pm 0.6$ mm and $5.6 \pm 0.8$ mm with a corresponding offset in the ZTD of $0.2 \pm 0.5$ mm, $-1.5 \pm 0.5$ mm, $-1.4 \pm 0.8$ mm, respectively. Similar values were obtained between solutions calculated for all stations/antennas for which individual calibration models are available. The corresponding offset in the ZTD has the opposite sign for the antennas with an offset in the up component larger than 5 mm (16 antennas) and, generally, does not exceed 2 mm. Such inconsistencies in the ZTD time series are not large enough to be captured during the combination process (see Section 3), where a 10 mm threshold in the ZTD bias (about 1.5 kg/m$^2$ IWV) is set in order to flag problematic ACs or stations.

**2.3    Impact of non-tidal atmospheric loading**

As reported in the International Earth Rotation and Reference Systems Service (IERS) Convention (2010), the diurnal heating of the atmosphere causes surface pressure oscillations with diurnal and semidiurnal variability and even higher harmonics. These atmospheric tides induce periodic

motions of the Earth's surface (Petrov and Boy, 2004). The conventional recommendation is to calculate the station displacement using the Ray and Ponte (2003) tidal model. However, crustal motion related to non-tidal atmospheric loading has been detected in station position time series from space geodetic techniques *(*van Dam et al., 1994; Magiarotti et al., 2001, Tregoning and Van Dam, 2005). Several models of station displacements related to this effect are currently available. Non-tidal atmospheric loading models are not yet considered as Class-1 models by the IERS (IERS 2010), indicating that there are currently no standard recommendations for data reduction. To evaluate their impact, two solutions, one with and one without a non-tidal atmospheric loading model, have been compared for the year 2013. In the solution with the model, the National Centers for Environmental Prediction (NCEP) model is used at the observation level during data reduction (Tregoning and Watson, 2009).

Dach et al. (2010) have already found that the repeatability of the station coordinates improves by 20% when applying the non-tidal atmospheric loading correction directly on the data analysis and by 10% when applying a post-processing correction to the resulting weekly coordinates. However, the effect on the ZTDs seems to be negligible. Generally, it causes a difference below 0.5 mm with a standard deviation not larger than 0.3 mm. The difference is thus below the level of confidence. Figure 4 shows time series of the differences of the ZTDs and the up components between two solutions obtained with and without non-tidal atmospheric loading for two EPN stations: KIR0 (Kiruna, Sweden) and RIGA (Riga, Latvia). Furthermore, there is no correlation between the values of estimated differences and vertical displacements caused by non-tidal atmospheric loading, as correlation coefficients for the analysed EPN stations were below 0.2.

## 3.   EPN-Repro2 combined solutions

The EPN ZTD combined product is obtained applying a generalized least square approach following the scheme described in Pacione et al. (2011). The first step in the combination process is the reading and checking of the SINEX TRO files delivered by the ACs. At this stage, gross errors (i.e. ZTD estimates with formal standard deviations larger than 15 mm) are detected and removed. The combination starts if at least three different solutions are available for a single site. Then, a first combination is performed to compute proper weights for each contributing solution, to be used in the final combination step. In this last step the combined ZTD estimates, their standard deviations and site/AC specific biases are determined. The combination fails if, after the first or second combination level, the number of ACs becomes less than three. Finally, ZTD site/AC specific biases exceeding 10 mm are investigated as potential outliers.

The EPN-Repro2 combination activities were carried out in two steps. First, a preliminary combined solution for the period 1996-2014 was performed taken all the available eight homogeneously reprocessed solutions (see Table 2) as input. The aim of this preliminary combined solution is to assess each contributing solution and to investigate site/AC specific biases prior to the final combination, flag the outliers and send feedback to the ACs. The agreement of each contributing solution w.r.t. the preliminary combination is given in terms of bias and standard deviation (not shown). The standard deviation is generally below 2.5 mm, with a clear seasonal behaviour (larger for larger ZTD values), while the bias is generally in the range of +/- 2 mm. However, there are several GPS weeks for which the bias and standard deviation exceeded the afore mentioned limits. To investigate these outliers, the time series of site/AC specific biases have been studied, since this analysis might be a useful tool to detect bad data periods and provide useful information for cleaning the EPN historical archive. An example is given in Figure 5 for the station VENE (Venice, Italy) for three contributing solutions AS0, GO4 and MU2 (G00 and GO1 are not shown but are very close to GO4). In the first years of the acquisition, the station VENE experienced tracking issues, clearly mirrored in both the bias and standard deviation time series.

All the site/AC specific biases are divided into three groups: the red group contains site/AC specific biases with values larger than 25 mm, the orange group contains site/AC specific biases in the range of [15 mm, 25 mm] and the yellow group contains site/AC specific biases in the range of [10 mm, 15 mm]. In Table 3 the percentages of red, orange and yellow biases for each contributing solution are summarized. The majority of biases belong to the yellow group; the percentage of biases in the orange group ranges from 12% for LP0 and LP1 solutions to 27% for the AS0 solution, while the percentage of biases in the red group ranges from 3% for the MU4 solution to 22% for the IG0 solution.

The final EPN-Repro2 tropospheric combination is based on the following input solutions: AS0, GO4, IG0, LP1 and MU2. MUT AC provided the MU2 solution after the preliminary combination, its only difference with respect to MU4 is the use of type mean antenna and individual calibration models, whose effect has already been described in section 2.2. For those ACs providing more than one solution, we have chosen the one carried out with the Vienna Mapping Function. The agreement in terms of bias and standard deviation of each contributing solution w.r.t. the final combination is shown in Figure 6.The standard deviation had improved significantly with respect to the preliminary combination (not shown here), due to the removal of outliers detected during this early combination. The standard deviation is below 3 mm before GPS week 1055 (26-03-2000) and

2 mm thereafter. This is related to the worse quality of data and products during the first years of the EPN/IGS activities.

The final EPN-Repro2 tropospheric combination is consistent with the final coordinate combination performed by the EPN Analysis Centre Coordinator. During the coordinate combination all stations were analyzed by comparing their coordinates for specific ACs and the preliminary combined values. In the cases where the differences were larger than 16 mm in the up component (vertical displacement), the station was eliminated and the whole combination process was repeated, up to three times, if necessary. This ensures the consistency of the individual contributing solution w.r.t. the final coordinates at the level of 16 mm in the up component. As internal quality metric, we have considered the site coordinate repeatability of the final coordinate combination (Figure 7). As a rule of thumb, 9 mm repeatability in the up component (i.e. 3 mm in ZTD as explained in Santerre, 1991) are needed to fulfill the requirement of retrieving IWV at an accuracy level of 0.5 kg/m2 (Bevis et al., 1994; Ning et al., 2016b). As shown in Figure 7, only at one site, MOPI (Modra Piesok, Slovakia), this threshold is exceeded on the long term. As reported at the EPN Central Bureau, MOPI has been excluded several times from the routine combined solutions because it has very bad observation periods in the past due to a radome manipulation that caused jumps in the up component. However, this 9 mm threshold has been temporary exceeded at several stations during bad periods, an example is given in Figure 8 for VENE (Venezia, Italy).

## 4.    Evaluation of the ZTD Combined Products with respect to independent data sets

The evaluation with respect to other sources or products, such as radiosonde data from the E-GVAP and numerical weather re-analysis from the European Centre for Medium-Range Weather Forecasts, ECMWF (ERA-Interim), provides a measure of the accuracy of the ZTD combined products.

### 4.1    Evaluation versus radiosonde

For the GNSS and radiosonde (RS) comparisons at the EPN collocated sites, we used profiles from the World Meteorological Organization (WMO) provided by EUMETNET in the framework of the Memorandum of Understanding between EUREF and EUMETNET. Radiosonde profiles are processed using a software by Haase et al. (2003) that checks the quality of the profiles, converts the dew point temperature to specific humidity, shifts the radiosonde profile to correct for the altitude offset between the GPS and the radiosonde sites, and determines the ZTD and IWV compensating for the change of the gravitational acceleration g with height.

A comparison of the GNSS and radiosonde ZTD time series for the EPN site CAGL (Cagliari, Sardinia Island, Italy) is shown in Figure 9, with the mean biases and standard deviations reported in the Figure. Similarly, we computed an overall bias (RS minus GNSS) and standard deviation for

all the 183 EPN collocated sites, using all the data available in the considered period (Figure 10). In this figure, the sites are sorted with increasing distance from the nearest radiosonde launch site. For instance, MALL (Palma de Mallorca, Spain) is the closest (0.5 km to the radiosonde site with WMO code 8301) while GRAZ (Graz, Austria) is the most distant (133 km to RS WMO code 14015). The amount of data available for the comparisons varies between sites, depending on the availability of the GNSS and radiosonde ZTD estimates in the considered epoch, and ranges from 121 pairs for VIS6 (Visby, Sweden, integrated in the EPN since 22-06-2014) up to 21226 pairs for GOPE (Ondrejov, Czech Republic, integrated in the EPN since 31-12-1995).

The mean relative [(RS-GNSS)/GNSS] bias ranges from -0.87%, which corresponds to -21.2 mm in ZTD (at EVPA, Ukraine, at a distance of 96.5 km from the RS WMO 33946 station) to 0.68%, which corresponds to 15.4 mm (at OBER, Germany at 90.8 km from RS WMO 11120). The overall mean ZTD bias for all sites is -0.6 mm (-0.03%) with a standard deviation of 4.9 mm (0.19%). For more than 75% of the stations (178 pairs), the agreement is below 5 mm in ZTD and only 5.5% of the stations (13 pairs) have ZTD biases higher than 10 mm. The higher biases arise mostly for paired sites over 50 km away from each other, for which differences in the geographical representativeness become important. For example, the GPS stations OBER, OBE2 and OBET located in Oberpfaffenhofen (Germany) are collocated with the RS WMO 11120 at Innsbruck Airport in Austria, on the opposite side of the North Chain in the Karwendel Alps. Our results are in accordance with Wang et al. (2007), in which the authors compared PW (not ZTD) from GPS and global radiosondes and reported an overall dry bias of about 1.08 mm for the radiosondes. However, it should be noted that these obtained biases, in both our and their study, are obtained from a mixture of radiosonde types, and daytime and nighttime RS launches. For instance, in agreement with Wang et al. (2007), we also found a small negative (dry) bias of -1.19 mm for Vaisala radiosondes (our bias is inversely calculated), which is the most common type used in Europe (81% of all used in this study). In this context, we mention that different Vaisala radiosonde types (e.g. RS80 vs RS90/RS92) are equipped with different humidity sensors, resulting in e.g. different RS-GNSS comparisons in PW, both for nighttime and daytime comparisons (e.g. Van Malderen et al., 2014). In addition, it must be kept in mind, that Wang et al. (2007) used global radiosonde data from 2003 and 2004, while we used all available data over Europe from 1994 to 2015. For MRZ, GRAW and M2K2 (from MODEM) radiosonde types, which represent 4.6%, 3.4% and 3.0% of the compared radiosondes types respectively, we received a systematic positive bias for the radiosondes, which can be interpreted as a moist bias, which is again in line with the results of Wang et al. (2007) for these radiosonde types. On the other hand, the results for M2K2 are at odds with Bock el al. (2013), in which a dry radiosonde bias in IWV compared to GPS was found at a French site.

However, they also indicated that their results are not consistent with another nearby radiosonde station and needs further investigation. Further investigation in our study is also needed for several near or moved GNSS stations, or switched radiosonde type at one station. For example in Brussels (Belgium) BRUS station, included in the EPN network since 1996, was replaced by BRUX in 2012. Their bias w.r.t. radiosonde (WMO code 6447) has opposite sign (-1.2 mm and 3.4 mm respectively). However, the radiosonde type was switched from RS80 to RS90 in 2007 (Van Malderen et al., 2014), which makes the bias for BRUS additionally affected by the change of the radiosonde type. .

In agreement with Ning et al. (2012), the ZTD standard deviation generally increases with the distance from the radiosonde launch site. It is in the range of [0.16; 0.76] %, which corresponds to [3; 18] mm in ZTD, till 15 km (first band in Figure 10); in [0.29; 0.78] %, corresponding to [7; 19] mm, till 70 km (second band in Figure 10), and in [0.43; 1.35] %, corresponding to [10; 33] mm till 133 km (third band in Figure 10). The numbers of the standard deviation are comparable with previous studies. Haase et al. (2001) showed a very good agreement with biases less than 5 mm in ZTD and a standard deviation of 12 mm for most of the analysed sites in the Mediterranean. Similar results (6.0 mm ± 11.7 mm) were obtained also by Vedel et al. (2001). Both studies were based on non-collocated pairs at sites less than 50 km from each other. Pacione et al (2011), considering 1-year of GPS ZTD and radiosonde data over the E-GVAP super sites network, obtained a standard deviation of 5-14 mm. Dousa et al. 2012 evaluated ZTDs from GNSS and radiosondes on a global scale over a 10-month period and reported a standard deviation of 5–16 mm.

If we compare both the EPN-Repro1 ZTD product (completed with the EUREF operational product after 30 December 2006) and the EPN-Repro2 with the radiosonde ZTDs for the same period 1996-2014, we found an improvement of approximately 3-4% in the overall standard deviation for the second processing.

**4.2    Evaluation versus ERA-Interim data**

We also compared the EPN-Repro2 ZTDs with the ZTDs calculated from ERA-Interim (Dee et al., 2011) from the European Centre for Medium-Range Weather Forecasts (ECMWF). The ERA-Interim is a re-analysis product of a Numerical Weather Prediction (NWP) model and is available every 6 hours (00, 06, 12, 18 UTC) with a horizontal resolution of 1×1 degree and with 60 vertical model levels.

For the period 1996-2014 and for each EPN station, the ZTD and tropospheric linear horizontal gradients were computed using the GFZ (German Research Centre for Geosciences) ray-tracing software (Zus et al., 2014). Combined EPN-Repro1 and EPN-Repro2 products as well as individual

ACs tropospheric parameters were assessed with the corresponding parameters estimated from the ERA-Interim re-analysis. The evaluation of GNSS and ERA-Interim was performed using the GOP-TropDB (Gyori and Dousa, 2016) by calculating parameter (ZTD, horizontal gradients, see below) differences for each station, using the values at every 6 hours (00:00, 06:00, 12:00 and 18:00), as available from the ERA-Interim model output. A linear temporal interpolation to those four timestamps was thus necessarily applied for all GNSS products, which are available in HH:30 timestamps as required for the combination process. As all compared GNSS products have the same time resolution (1 hour), the interpolation is assumed to affect all products in the same way. Therefore, we assume that all inter-comparisons to a common reference (ERA-Interim) principally reflect the quality of the products. No vertical corrections were applied since ERA-Interim variables were estimated for the long-term antenna reference position of each station.

Table 4 summarizes the mean total statistics of individual (ACs) and combined (EUREF) tropospheric parameters, ZTDs and horizontal gradients, over all available stations. The EUREF combined solution does not provide tropospheric gradients and these could therefore be evaluated for individual solutions only. In Table 4, a common ZTD bias (ERA-Interim minus GNSS) of about 1.8 mm is found for all GNSS solutions compared to ERA-Interim, but a large station to station variability could be noted, as is obvious from the estimated uncertainties. ZTD standard deviations are generally at the level of 8 mm between GNSS and ERA-Interim ZTDs, but with the IG0 solution performing about 25% worse than the others as already detected during the combination. Two solutions, AS0 and LP1 are slightly better than GO4 and MU2: with a standard deviation of 7.7 mm, their accuracy is at the level of the EUREF combined solution. The better performance of the AS0 solution can be explained by applying a stochastic troposphere modelling using original (not double-difference) observations sensitive to the absolute tropospheric delays, so that the true dynamics in the troposphere is better taken into account. LP1 included roughly one third of the EPN stations, properly selected according to the station quality, hereby making it difficult to interpret this difference with respect to those solutions processing the full EPN.

The comparison of tropospheric linear horizontal gradients (East and North) from GNSS and ERA-Interim revealed a problem with the MU2 solution (see Table 4). This solution shows a high inconsistency over different stations, which is not visible in the total statistics, but mainly in the uncertainties, which are an order of magnitude higher compared to all other solutions. A geographical plot (not shown here) confirmed this site-specific systematic effect, both in positive and negative sense. The impact was however not observed in the MU2 ZTD results. Additionally, the GO4 solution performed slightly worse than the others. This was identified as a consequence of

estimating 6-hour gradients using a piece-wise linear function without any absolute or relative constraints. In such case, higher correlations with other parameters occurred and increased the uncertainties of the estimates. For this purpose, the GO6 solution (not shown) was derived, fully compliant with the GO4, but stacking tropospheric gradients into 24 hours piece-wise linear modelling. In comparison with the former GO4 solution (Dousa and Vaclavovic, 2016), the GO6 standard deviations dropped from 0.38 mm to 0.28 mm and from 0.40 mm to 0.29 mm for East and North gradients, respectively, which corresponds to the LP1 solution that applied the same settings. Additionally, Dousa and Vaclavovic (2016) found a strong impact of a low-elevation receiver tracking problem on the estimation of the horizontal gradients, which was particularly visible when comparing with ERA-Interim horizontal gradients. Looking for systematic behaviour in monthly mean differences in the gradients therefore seems to be a useful indicator for instrumentation-related issues and should be applied as one of the tools for cleaning the EPN historical archive.

For completeness, we also evaluated the EPN-Repro1 ZTD product with respect to ERA-Interim using the same period, i.e. 1996-2014 (after completing again with the EUREF operational product, see above). Comparing EPN-Repro1 and EPN-Repro2 with the numerical weather model re-analysis showed an 8-9% improvement of EPN-Repro2 in both overall standard deviation and bias. Figure 11 shows the distributions of station mean biases and standard deviations of EPN-Repro1 and EPN-Repro2 ZTDs compared to ERA-Interim ZTDs using the whole period 1996-2014. Common reductions of both statistical characteristics are clearly visible for the majority of all stations. From the data of Figure 11, we also illustrate the site-by-site improvements in terms of ZTD bias, standard deviation and RMS in Figure 12. The calculated median improvements for these statistics reached 21.1 %, 6.8 % and 8.0 %, respectively, which corresponds to the above mentioned improvement of 8-9 %. A degradation of the standard deviation was found at three stations: SKE8 (Skellefteaa, Sweden, integrated in the EPN since 28-09-2014), GARI (Porto Garibaldi, Italy, integrated in the EPN since 08-11-2009) and SNEC (Snezka, Czech Republic, former EPN station since 14-06-2009). These three stations provide much less data compared to other stations, respectively only 1%, 30% and 3% of data pairs available at other stations. All other stations (290) showed improvements. We found 72 stations with increased absolute bias in EPN-Repro2 compared to EPN-Repro1 while the other 221 stations (75%) had a reduced bias with ERA-Interim ZTD.

Time series of monthly mean biases and standard deviations for ZTD differences of EPN-Repro2 and ERA-Interim are shown in Figure 13. The small negative bias slowly decreases towards 2014, but the high uncertainty of the mean bias indicates a site-specific behaviour, depending mainly on

latitude and altitude of the EPN station and the quality of both ERA-Interim and GNSS products. There is almost no seasonal signal observed in the time series of ZTD mean biases or uncertainties, but clearly in the ZTD mean standard deviation and the uncertainties. The increase of standard deviation in summer is due to more humidity in troposphere which is more difficult to model accurately in both GNSS and ERA-interim. The slightly increasing standard deviation towards 2014 can be attributed to the increase of number of stations in EPN: starting from about 30 in 1996 and with more than 250 in 2014. A higher number of stations reduces the variability in monthly mean biases, however, site-specific errors then contribute more to higher values of standard deviation.

Figure 14 displays the geographical distribution of total ZTD biases (ERA-Interim minus GNSS) and standard deviations for all sites. Prevailing positive biases seem to become lower or even negative in the mountain areas. There is no latitudinal dependence observed for ZTD biases in Europe, but a strong one for standard deviations. This corresponds mainly to the increase of water vapour content and its variability towards the equator.

**4.3 Evaluation of ZTD trends**

To illustrate the impact of the new processing on the resulting ZTD trends and related uncertainties, we considered five EPN stations, among those with the longest time span: GOPE (Ondrejov, Czech Republic, integrated in the EPN since 31-12-1995), METS (Kirkkonummi, Finland, integrated in the EPN since 31-12-1995), ONSA (Onsala, Sweden, integrated in the EPN since 31-12-1995), PENC (Penc, Hungary, integrated in the EPN since 03-03-2096) and WTZR (Bad Koetzting, Germany, integrated in the EPN since 31-12-1995). For these five stations, we have computed ZTD trends using EPN-Repro2, EPN-Repro1 (again completed with the EUREF operational products), radiosonde and ERA-Interim data. Furthermore, those five stations also belong to the IGS Network, for which IGS Repro1, completed with the IGS operational products, are available and extracted from the GOP-TropDB, so that we could also calculate ZTD trends from this dataset.

First, we removed the annual signal from the original time series and marked all outliers according to the 3-sigma criterion. Then, we tried to remove all inhomogeneities in the GNSS ZTD time series, related to instrumental changes, which might introduce a change in the mean of the ZTD time series and therefore have an impact on the ZTD trends. In particular, for all GNSS ZTD data sets we have estimated all documented shifts in the mean related to the antenna replacement. No other unexplained break points has been corrected for, to be sure not to introduce any artificial errors. Based on these cleaned and filtered data, we have used, independently, a linear regression model before and after the considered epoch of the offset. The difference of the mean ZTDs between those two linear regression models is then considered as the offset of the specific epoch. With this

technique, we removed all the estimated offsets from the original GNSS ZTD time series. Generally, the amplitudes of the offsets are much lower than the noise level and depend on the applied method of estimation. Therefore, the final ZTD trends and uncertainties presented here are affected by the used methodology and should not be considered in absolute terms. No homogenization has been done for the radiosonde data, since reliable metadata are not available. Also the ERA-Interim ZTD time series were not corrected for inhomogeneities. Finally, a Least Squares Estimation method has been applied to estimate the linear trends and the seasonal components.

In Figure 15, the ZTD trends and uncertainties are presented for the five sites and for all ZTD datasets. First of all, it should be noted that the trends between the three GNSS ZTD data sets are very consistent (as long as the same homogenisation procedure is applied). The overall RMS among trends estimated from GNSS measurements is 0.02 mm/year. If we now consider all five ZTD sources, the best agreement between the ZTD trends is achieved at ONSA (RMS = 0.04 mm/year) and WTZR (RMS = 0.02 mm/year). For PENC, we also have a good agreement of the GNSS ZTD trends with respect to ERA-Interim (RMS = 0.05 mm/year), but a large discrepancy with the radiosonde ZTD trend is found (RMS = -0.31 mm/year). This large discrepancy is probably due to the distance to the radiosonde launch site (40.7 km, RS WMO 12843) and to the lack of homogenization of the radiosonde data. For the five considered stations, the agreement of GNSS ZTD trends with respect to ERA-Interim (RMS = 0.11 mm/year) is better than with respect to radiosondes (RMS = 0.16 mm/year). Even although, for the five considered stations, EPN-Repro2 do not change significantly the value of the ZTD trends with respect to EPN-Repro1, it has a less uncertainty (the improvement is 6.9%) of ZTD trends, better agreement with ERA-Interim (the improvement is 8.0%) ZTD trends and a slightly worse agreement with the radiosonde (the degradation is 3.8%). However, one should keep in mind that time series from radiosonde measurements were not homogenized and their trends may not be necessarily trustworthy. Over Europe, the EPN network has a better spatial resolution than the IGS and radiosonde networks, which are used today for an observations-based long-term analysis of ZTD/IWV variability. Taking into account the good consistency among the ZTD trends, EPN-Repro2 can be used for trend detection in areas where other data are not available.

## 5.  Conclusions

In this paper, we described the activities carried out in the framework of the second reprocessing campaign of the EPN. We focused on the tropospheric products homogenously reprocessed by five EPN Analysis Centres for the period 1996-2014 and we described the ZTD combined product. We evaluated the impact of few diversities among the provided GNSS solutions. The inclusion of

additional GLONASS observations in the GNSS processing has a neutral impact on the ZTD trend analysis pointing out that the ZTD trends might be determined independently of the satellite systems used in the processing (see Section 2.1). The inconsistencies in the ZTD time series due to different antenna calibration models (see Section 2.2) are not large enough to be captured during the combination process (see Section 3), where a 10 mm threshold in the ZTD bias (about 1.5 kg/m$^2$ IWV) is set in order to flag problematic ACs or stations. The effect on the ZTDs of non-tidal atmospheric loading correction (see Section 2.3) seems to be negligible. We assessed the quality of the ZTD combined product, which is below 3 mm before GPS week 1055 (26-03-2000) and 2 mm thereafter. This is related to the worse quality of data and products during the first years of the EPN/IGS activities.

Both individual and combined tropospheric products, along with reference coordinates and other metadata, are stored in a SINEX TRO format (Gendt, G. 1997), and are available to the users at the EPN Regional Data Centres (RDC), located at BKG (Federal Agency for Cartography and Geodesy, Germany). For each EPN station, plots on ZTD time series, ZTD monthly means, comparison with radiosonde data (if collocated), and comparison versus the ERA-Interim data will be available at the EPN Central Bureau (Royal Observatory of Belgium, Brussels, Belgium).

We showed in section 4.1 that EPN-Repro2 led to an improvement of approximately 3-4% in the overall standard deviation in the ZTD differences with radiosonde data, as compared with EPN-Repro1.

The assessment of the EPN-Repro2 comparison with the ERA-Interim re-analysis showed an 8-9% improvement in both the overall ZTD bias and standard deviation with respect to EPN-Repro1 for the majority of the stations (see Section 4.2). Comparisons of the GNSS solutions with ERA-Interim, showed the agreement in ZTD at the level of 8-9 mm, however, site performance ranging from 5 mm to 15 mm for standard deviations and from -7 mm to 3 mm for biases when neglecting outliers (<1%).

The use of ground-based GNSS long-term data for climate research is an emerging field. For example, for the assessment of Euro-CORDEX (Coordinated Regional Climate Downscaling Experiment) climate model simulation, the IGS Repro1dataset (Byun and Bar-Sever, 2009) has been used as reference reprocessed GPS products (Bastin et al. 2016). However, this dataset is quite sparse over Europe (only 85 stations over the 280 EPN stations) and covers only the period 1996-2010. As pointed by Baldysz et al. (2015, 2016) an additional two years of ZTD data can change the estimated trends up to 10%. Therefore, with data after 2010 and with a better coverage over Europe, EPN-Repro2 can be used as a reference data set with a high potential for monitoring the trends and

variability in atmospheric water vapour as reported in Section 4.3. As a matter of fact, a comparison between GNSS IWV, computed from EPN-Repro2 ZTD data for SOFI (Sofia, Bulgaria) by the Sofia University, and ALADIN-Climate IWV simulations conducted by the Hungarian Meteorological Service, is performed for the period 2003-2008 at the moment. The preliminary results show a tendency of the model to underestimate IWV. Clearly, a larger number of model grid points needs to be investigated in different regions in Europe and the EPN-Repro2 data is well suited for this.

The reprocessing activity of the five EPN ACs was a huge effort generating homogeneous products not only for station coordinates and velocities, but also for tropospheric products. The knowledge gained will certainly help for a next reprocessing activity. A next reprocessing will most likely include Galileo and BeiDou data and therefore it will be started in some years from now after having successfully integrated these new data in the current operational near real-time and daily products of EUREF. The consistent use of identical models in various software packages is another challenge for the future and would enable to improve the consistency of the combined solution. Prior to any next reprocessing, it was agreed in EUREF to focus on cleaning and documenting the data in the EPN historical archive as it should highly facilitate any future work. For this purpose, all existing information needs to be collected from all the levels of data processing, combination and evaluation, which includes initial GNSS data quality checking, generation of individual daily solutions, combination of individual coordinates and ZTDs, long-term combination for velocity estimates and assessments of ZTDs and gradients with independent data sources.

*Author Contributions.* R. Pacione coordinated the writing of the manuscript and wrote section 1, 2, 3 and 4.1. A. Araszkiewicz wrote section 2.2 and 2.3, 4.3 and contributed to section 4.1. E. Brockmann wrote section 2.1. J. Dousa wrote section 4.2. All authors contributed to section 5. All authors approved the final manuscript before its submission.

**Acknowledgments**

The authors would like to acknowledge the support provided by COST – (European Cooperation in Science and Technology) for providing financial assistance for the publication of the paper. The authors thank the members of the EUREF project "EPN reprocessing". e-GEOS work is done under ASI Contract 2015-050-R.0. The assessments of the EUREF combined and individual solutions in the GOP-TropDB were supported by the Ministry of Education, Youth and Science, the Czech Republic (project LH14089). The MUT AC contribution was supported by statutory

founds at the Institute of Geodesy, Faculty of Civil Engineering and Geodesy, Military University of Technology (No. PBS/23-933/2016). Finally, we thank the two anonymous referees and the Associate Editor Dr. Roeland Van Malderen for their comments which helped much to improve the paper.

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

**Table**

**Table Captions**

Table 1: EPN Analysis Centres providing EPN-Repro2 solutions**.**

Table 2: EPN-Repro2 processing options for each contributing solutions. AS0 solution is provided by ASI/CGS (Matera, Italy), GO0, GO1 and GO4 solutions are provided by GOP (Pecny, Czech Republic), IG0 solution by IGE (Madrid, Spain), LP0 and LP1 solutions by LPT (Waben, Switzerland), and MU2 and MU4 solutions by MUT (Warsaw, Poland). (PPP=Precise Point Positioning; EGM=Earth Gravitational Model, GMF=Global Mapping Function; VMF=Vienna Mapping Function; HOI=Higher Order Ionosphere, IONEX=IONospheric maps Exchange format, IGRF=International Geomagnetic Reference Field, FES=Finite Element Solution).

Table 3. Percentage of red, orange and yellow biases (see text) for each contributing solution**.**

Table 4. Mean statistics and uncertainties, calculated from results of individual stations, provided for AC individuals and EUREF combined (EPN-Repro1 and EPN-Repro2) tropospheric parameters compared to the ERA-Interim re-analysis (ERA-Interim minus GNSS). EGRD represents east gradient and NGRD north gradient.

| AC | Full name | City | Country | SW | EPN Network |
|---|---|---|---|---|---|
| ASI | Agenzia Spaziale Italiana | Matera | Italy | GIPSY-OASIS II | Full EPN |
| GOP | Geodetic Observatory | Pecny | Czech Republic | Bernese | Full EPN |
| IGE | National Geographic Institute | Madrid | Spain | Bernese | EPN-Subnetwork |
| LPT | Federal Office of Topography | Wabern | Switzerland | Bernese | EPN-Subnetwork |
| MUT | Military University of Technology | Warsaw | Poland | GAMIT | Full EPN |

Table 1: EPN Analysis Centres providing EPN-Repro2 solutions.

| | AS0 | GO0 | GO1 | GO4 | IG0 | LP0 | LP1 | MU2 | MU4 |
|---|---|---|---|---|---|---|---|---|---|
| **SOFTWARE** | GIPSY 6.2 | Bernese 5.2 | | | Bernese 5.2 | Bernese 5.2 | | GAMIT 10.5 | |
| **GNSS** | GPS | GPS | | | GPS + GLONASS | GPS+GLONASS | | GPS | |
| **SOLUTION TYPE** | PPP | Network | | | Network | Network | | Network | |
| **STATIONS** | Full EPN | Full EPN | | | EPN Subnetwork | EPN Subnetwork | | Full EPN | |
| **ORBITS** | JPL Repro2 | CODE Repro2 | | | CODE Repro2 | CODE Repro2 | | CODE Repro2 | |
| **ANTENNAS** | IGS08 | IGS08 + Individual. | | | IGS08+ Individual. | IGS08 | IGS08 + Individual. | IGS08 + Individual | IGS08 |
| **IERS** | 2010 | 2010 | | | 2010 | 2010 | | 2010 | |
| **GRAVITY** | EGM08 | EGM08 | | | EGM08 | EGM08 | | EGM08 | |
| **TROPOSPHERE Estimated Parameters** | ZTD (5min) GRAD (5min) | ZTD (1h) GRAD (6h) | | | ZTD (1h) GRAD (6h) | ZTD (1h) GRAD (24h) | | ZTD (1h) GRAD (24h) | |
| **MAPPING FUNCTION** | VMF | GMF | VMF1 | VMF | GMF | GMF | VMF | VMF | |
| **ZTD/GRAD time stamp** | hh:30 24 estimates/day | hh:30 (and hh:00) 24(+24) estimates/day | | | hh:30 24 estimates/day | hh:30 (and hh:00) 24(+24) estimates/day | | hh:30 24 estimates/day | |
| **IONOSPHERE** | HOI included | CODE, HOI included | | | CODE (HOI included) | CODE (HOI included) | | CODE IONEX + IGRF11 (HOI included) | |
| **REFERENCE. FRAME** | IGb08 | IGb08 | | | IGb08 | IGb08 | | IGb08 | |
| **OCEAN TIDES** | FES2004 | FES2004 | | | FES2004 | FES2004 | | FES2004 | |
| **TIDAL-ATMOSPHERIC LOADING** | NO | NO | | | YES | YES | YES | YES | |
| **NON-TIDAL-ATMOSPHERIC LOADING** | NO | NO | NO | YES | NO | NO | YES | NO | |
| **ELEVATION CUTOFF** | 3 | 3 | | | 3 | 3 | | 5 | |
| **Delivered SNX_TRO Files [from week to week]** | 0834-1824 | 0836-1824 | | | 0835-1816 | 0835-1802 | | 0835-1824 | |

Table 2: EPN-Repro2 processing options for each contributing solutions. AS0 solution is provided by ASI/CGS (Matera, Italy), GO0, GO1 and GO4 solutions are provided by GOP (Pecny, Czech Republic), IG0 solution by IGE (Madrid, Spain), LP0 and LP1 solutions by LPT (Waben, Switzerland), and MU2 and MU4 solutions by MUT (Warsaw, Poland). (PPP=Precise Point Positioning; EGM=Earth Gravitational Model, GMF=Global Mapping Function; VMF=Vienna Mapping Function; HOI=Higher Order Ionosphere, IONEX=IONospheric maps Exchange format, IGRF=International Geomagnetic Reference Field, FES=Finite Element Solution).

| Solution | %Red bias | % Orange bias | % Yellow bias |
|----------|-----------|---------------|---------------|
| AS0 | 17 | 27 | 56 |
| G00 | 10 | 22 | 67 |
| G01 | 12 | 23 | 65 |
| G04 | 12 | 23 | 65 |
| IG0 | 22 | 14 | 64 |
| LP0 | 10 | 12 | 79 |
| LP1 | 10 | 12 | 78 |
| MU2 | 3 | 15 | 82 |

Table 3. Percentage of red, orange and yellow biases (see text) for each contributing solution.

| Solution | ZTD bias [mm] | ZTD sdev [mm] | EGRD bias [mm] | EGRD sdev [mm] | NGRD bias [mm] | NGRD sdev [mm] |
|---|---|---|---|---|---|---|
| AS0 (full EPN) | 1.7±2.0 | 7.7±1.9 | -0.00±0.06 | 0.32±0.09 | -0.09±0.06 | 0.33±0.10 |
| GO4 (full EPN) | 1.9±2.4 | 8.1±2.1 | 0.04±0.09 | 0.38±0.10 | -0.00±0.09 | 0.40±0.12 |
| MU2 (full EPN) | 1.8±2.0 | 8.3±2.1 | 0.03±0.32 | 0.35±2.46 | 0.01±0.84 | 0.34±2.37 |
| IG0 (part EPN) | 1.6±2.3 | 10.7±2.2 | 0.05±0.09 | 0.33±0.11 | -0.04±0.12 | 0.36±0.12 |
| LP1 (part EPN) | 1.7±2.4 | 7.7±1.7 | 0.02±0.06 | 0.28±0.05 | -0.03±0.09 | 0.27±0.06 |
| EPN-Repro2 | 1.8±2.1 | 7.8±2.2 | - | - | - | - |
| EPN-Repro1 | 2.2±2.3 | 8.5±2.1 | - | - | - | - |

Table 4. Mean statistics and uncertainties, calculated from results of individual stations, provided for AC individuals and EUREF combined (EPN-Repro1 and EPN-Repro2) tropospheric parameters compared to the ERA-Interim re-analysis (ERA-Interim minus GNSS). EGRD represents east gradient and NGRD north gradient.

**Figure**

**Figure Captions**

Figure 1. Time series of the number of GNSS observations for the period 1996-2014. GPS observations are shown in red, GPS+GLONASS in blue and their differences in green. The difference becomes significant starting from 2008.

Figure 2. ZTD trend differences between GPS only and GPS+GLONASS, computed over 111 sites. The rate is in violet (primary y-axis) and the number of used differences is in green (secondary y-axis).

Figure 3. EPN station KLOP (Kloppenheim, Frankfurt, Germany) ZTD differences time series between solutions processed with 'individual' and 'type mean' antenna calibration models. Two instrumentation changes occurred at the station (marked by vertical dashed red lines): the first in June 27th 2007, when the previous antenna was replaced with a TRM55971.00 and a TZGD radome, and the second in June 28th 2013 with the installation of a TRM57971.00 and a TZGD radome.

Figure 4. Left part: Time series of the ZTD and up component differences between two time series obtained with and without non-tidal atmospheric loading for two EPN stations: KIR0 (Kiruna, Sweden) and RIGA (Riga, Latvia). Right part: Scatter plots between these two parameters.

Figure 5: VENE (Venice, Italy) time series of ZTD biases and standard deviations for the three contributing solutions AS0, GO4 and MU4 with respect to the combined solution for the period July 21st, 1996 - July 28, 2007 (GPS weeks 0863-1437). GO0 and GO1 are not shown here, since they are very close to GO4.

Figure 6: Weekly mean ZTD biases (upper part) and standard deviations (lower part) of each contributing solution w.r.t. the final EPN-Repro2 combination.

Figure 7. Long term up component repeatability of the final coordinates for all stations. The site coordinate repeatability is used as an internal quality metric. Stations are sorted by name.

Figure 8 VENE (Venice Italy) time series of daily repeatability (for definition, see Figure. 7) in the up component for the period July 21st, 1996 - July 28, 2007 (GPS weeks 0863-1437).

Figure 9 EPN station CAGL (Cagliari, Sardinia Island, Italy). Upper part: Radiosondes (in red) and GPS (in blue) ZTD time series. Lower part: ZTD differences, calculated as RS minus GNSS.

Figure 10: RS minus GNSS ZTD biases for all GNSS-RS station pairs. The error bar is the standard deviation. Sites are sorted with increasing distances from the nearest radiosonde launch site.

Figure 11: Distributions of station mean ERA-Interim minus GNSS ZTD biases (left) and standard deviations (right) of EPN-Repro1 and Repro2 compared to ERA-Interim.

Figure 12: Site-by-site ZTD improvements of EPN-Repro2 versus EPN-Repro1 compared to ERA-Interim

Figure 13: Time series of monthly mean biases (lower part) and standard deviations (upper part) for ZTD differences between EPN-Repro2 and ERA-Interim re-analysis (ERA-Interim minus GNSS). Uncertainties are calculated over all stations.

Figure 14: Geographical distribution of ZTD biases (left) and standard deviations (right) for EPN-
Repro2 compared to ERA-Interim ( ERA-Interim minus GNSS.

Figure 15: ZTD trend comparisons at five EPN stations for 5 different ZTD datasets. The error bars
are the formal errors of the estimated trend values.

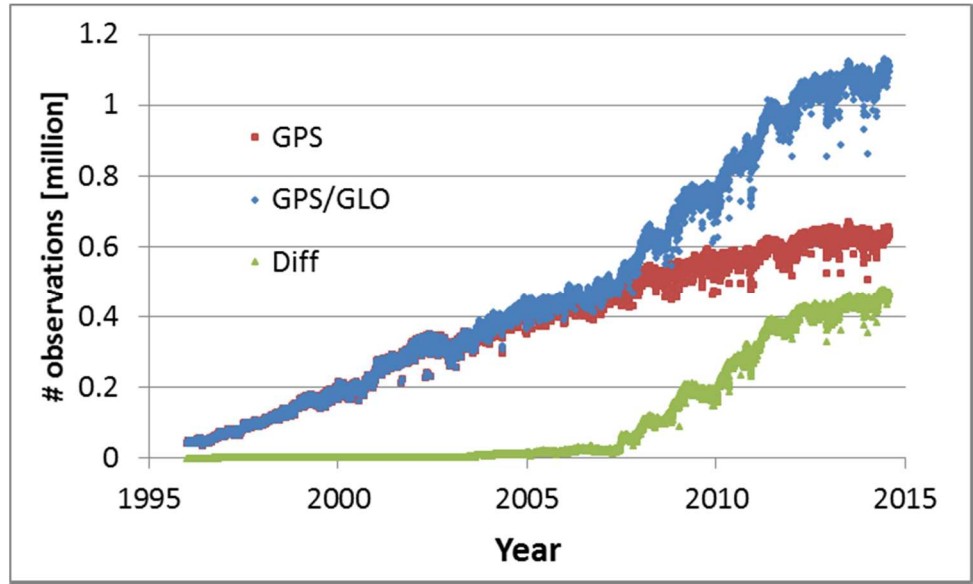

Figure 1. Time series of the number of GNSS observations for the period 1996-2014. GPS
observations are shown in red, GPS+GLONASS in blue and their differences in green. The
difference becomes significant starting from 2008.

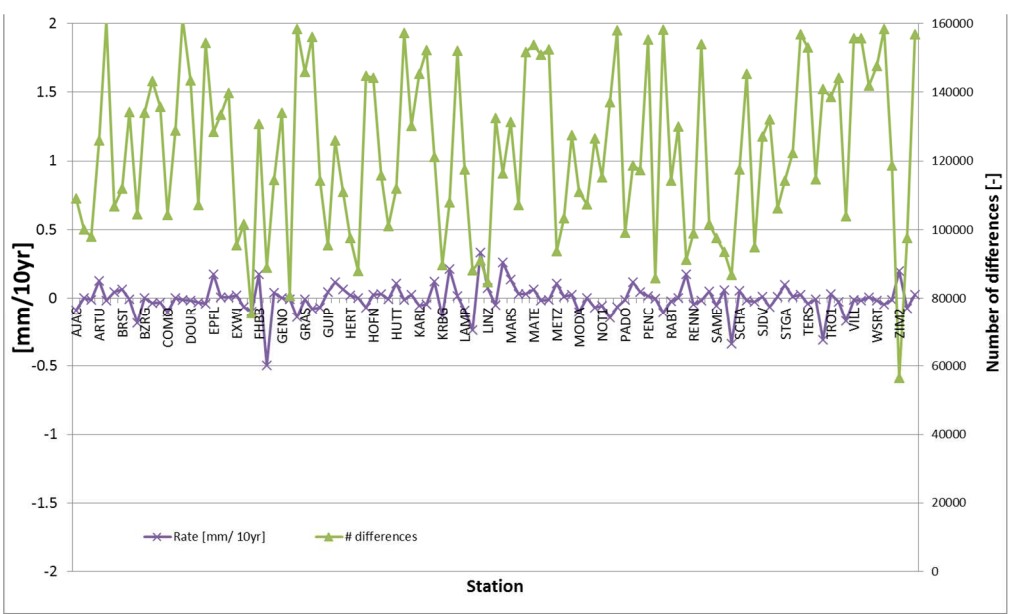

Figure 2. ZTD trend differences between GPS only and GPS+GLONASS, computed over 111 sites.
The rate is in violet (primary y-axis) and the number of used differences is in green (secondary y-
axis).

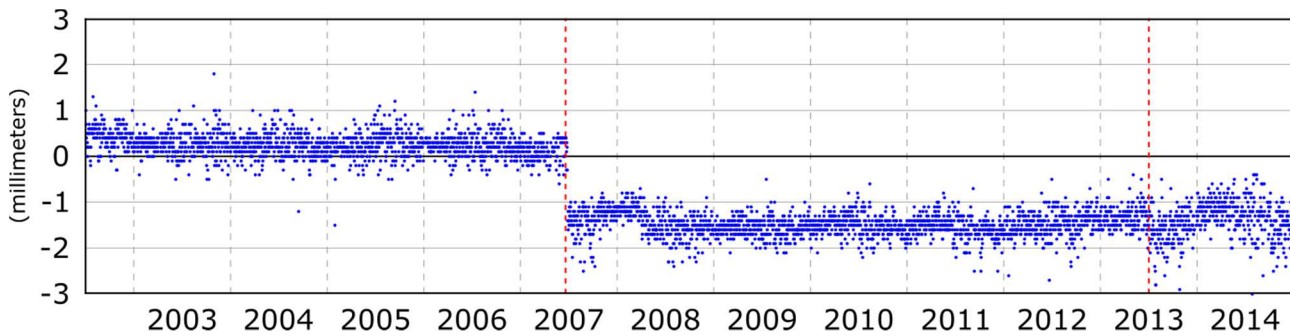

Figure 3. EPN station KLOP (Kloppenheim, Frankfurt, Germany) ZTD differences time series
between solutions processed with 'individual' and 'type mean' antenna calibration models. Two
instrumentation changes occurred at the station (marked by vertical dashed red lines): the first in
June 27th 2007, when the previous antenna was replaced with a TRM55971.00 and a TZGD radome,
and the second in June 28th 2013 with the installation of a TRM57971.00 and a TZGD radome.

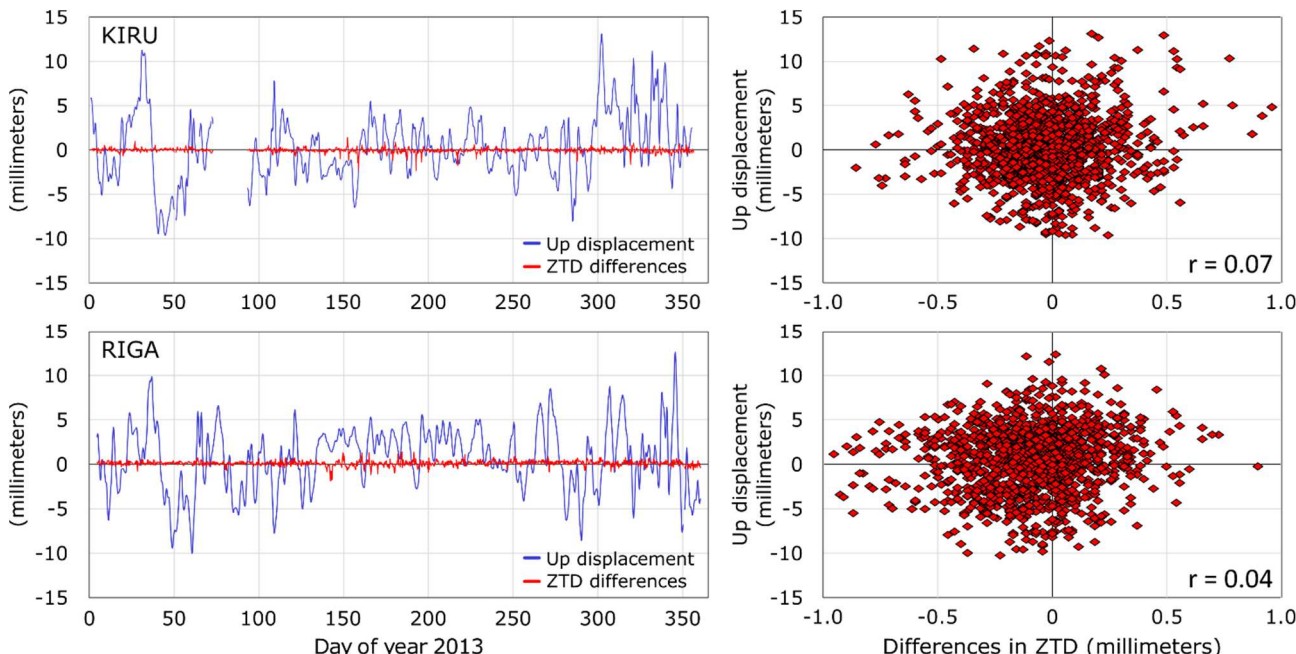

Figure 4. Left part: Time series of the ZTD and up component differences between two time series obtained with and without non-tidal atmospheric loading for two EPN stations: KIR0 (Kiruna, Sweden) and RIGA (Riga, Latvia). Right part: Scatter plots between these two parameters.

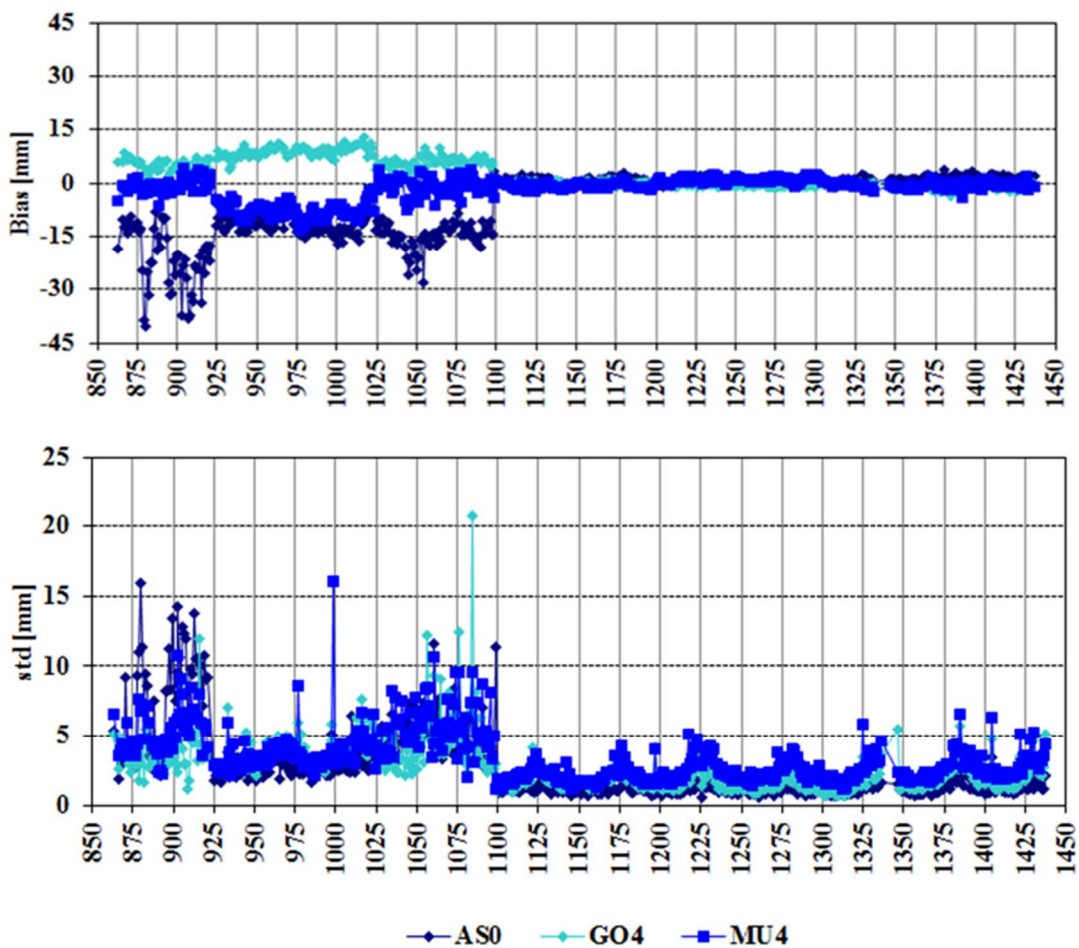

Figure 5: VENE (Venice, Italy) time series of ZTD biases and standard deviations for the three contributing solutions AS0, GO4 and MU4 with respect to the combined solution for the period July 21st, 1996 - July 28, 2007 (GPS weeks 0863-1437). GO0 and GO1 are not shown here, since they are very close to GO4.

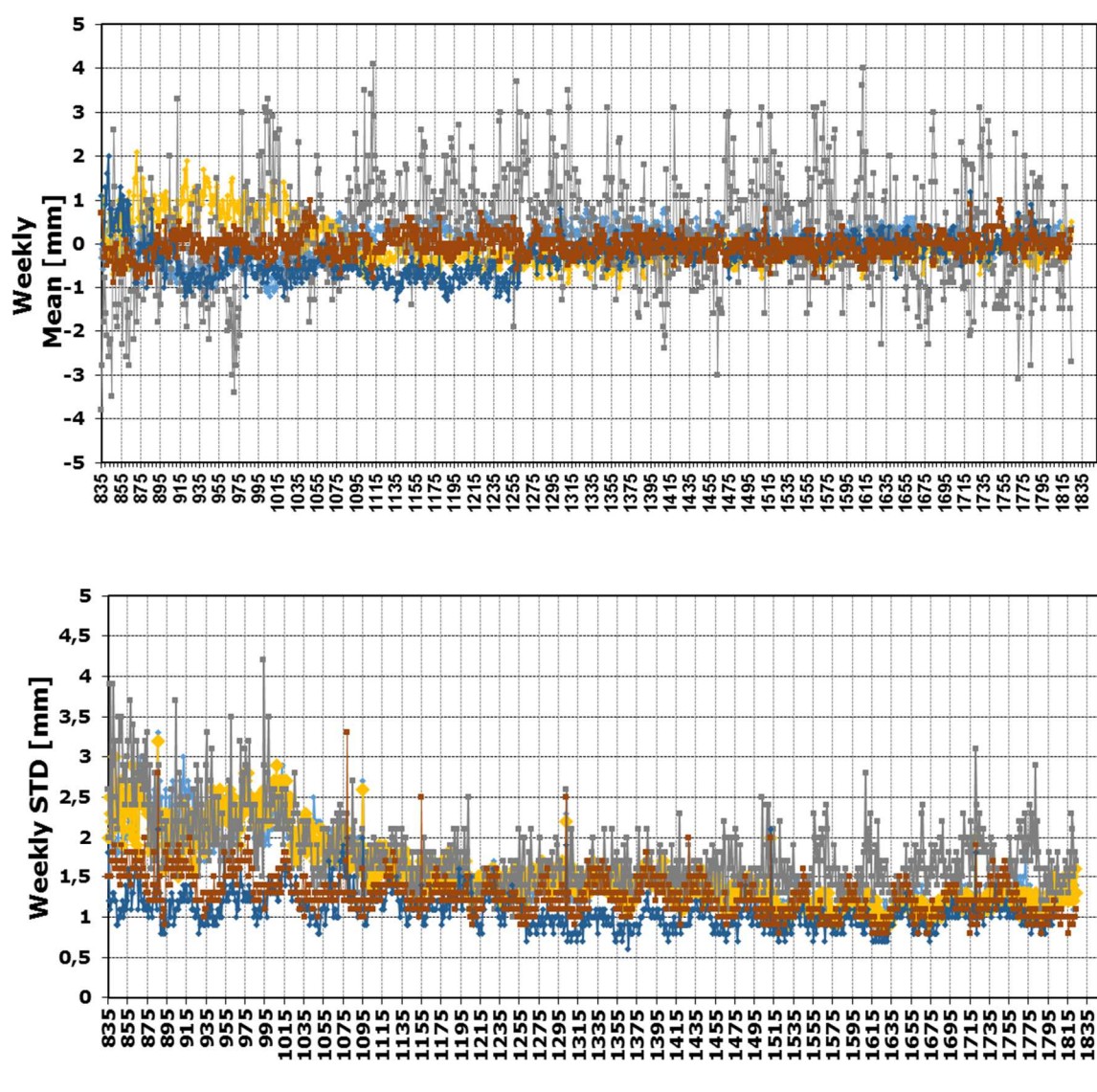

Figure 6: Weekly mean ZTD biases (upper part) and standard deviations (lower part) of each contributing solution w.r.t. the final EPN-Repro2 combination.

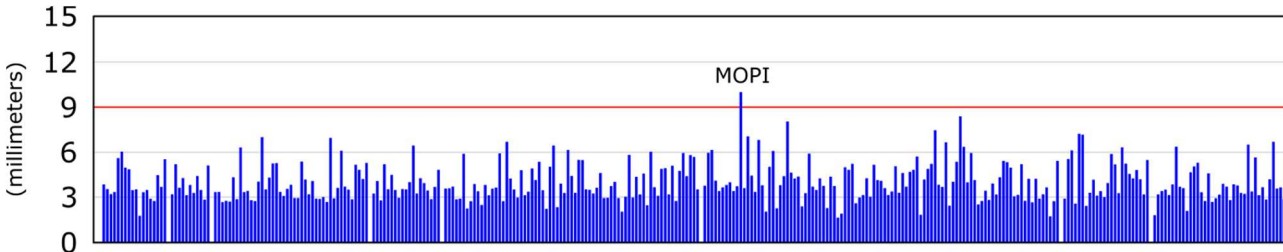

Figure 7. Long term up component repeatability of the final coordinates for all stations. The site coordinate repeatability is used as an internal quality metric. Stations are sorted by name.

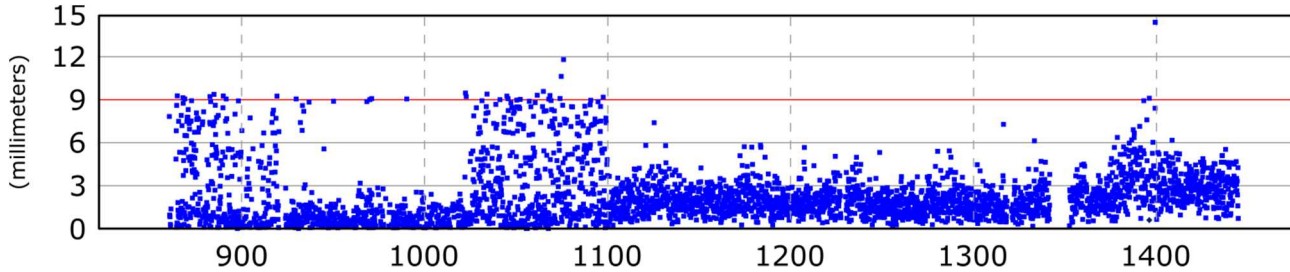

Figure 8 VENE (Venice Italy) time series of daily repeatability (for definition, see Figure. 7) in the up component for the period July 21$^{st}$, 1996 - July 28, 2007 (GPS weeks 0863-1437).

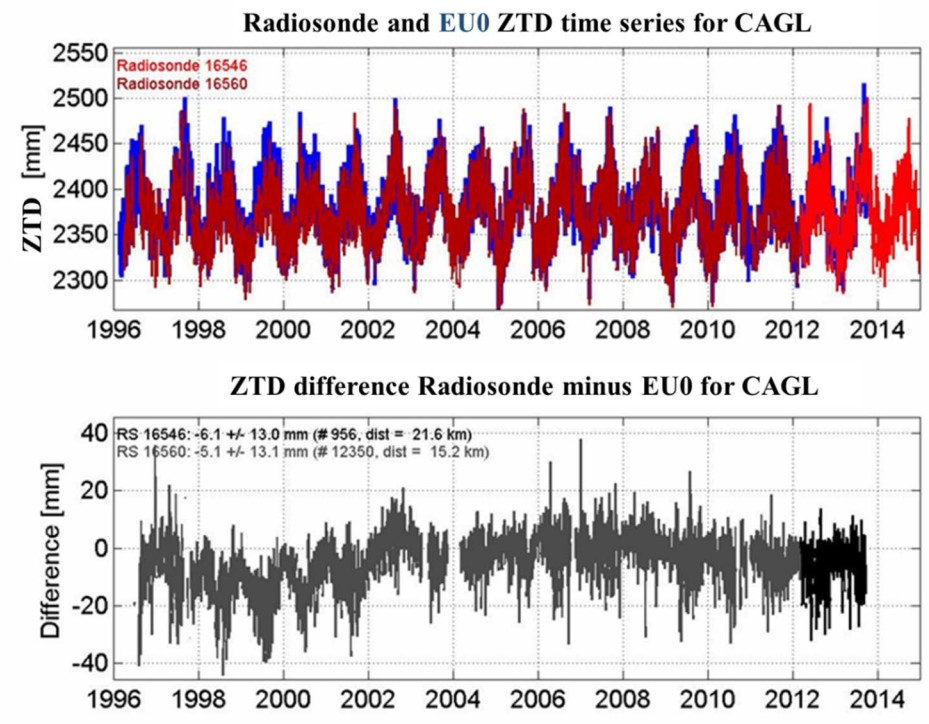

Figure 9 EPN station CAGL (Cagliari, Sardinia Island, Italy). Upper part: Radiosondes (in red) and
GPS (in blue) ZTD time series. Lower part: ZTD differences, calculated as RS minus GNSS.

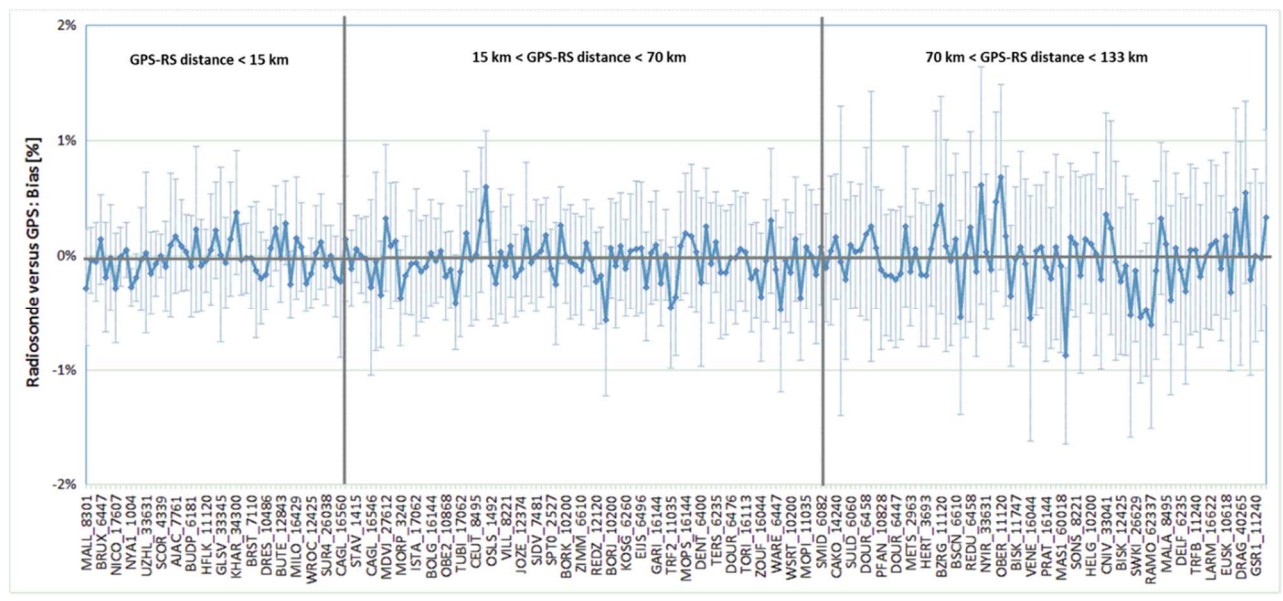

Figure 10: RS minus GNSS ZTD biases for all GNSS-RS station pairs. The error bar is the standard deviation. Sites are sorted with increasing distances from the nearest radiosonde launch site. The x-axis shows the GNSS station and the radiosonde site WMO code.

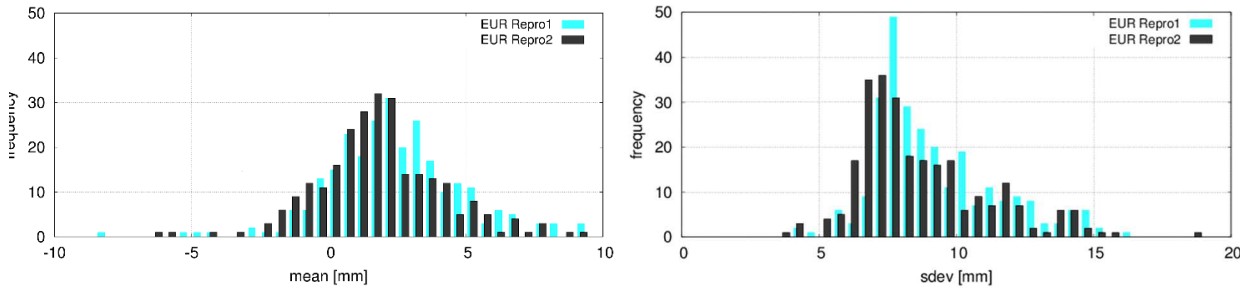

Figure 11: Distributions of station mean ERA-Interim minus GNSS ZTD biases (left) and standard
deviations (right) of EPN-Repro1 and Repro2 compared to ERA-Interim.

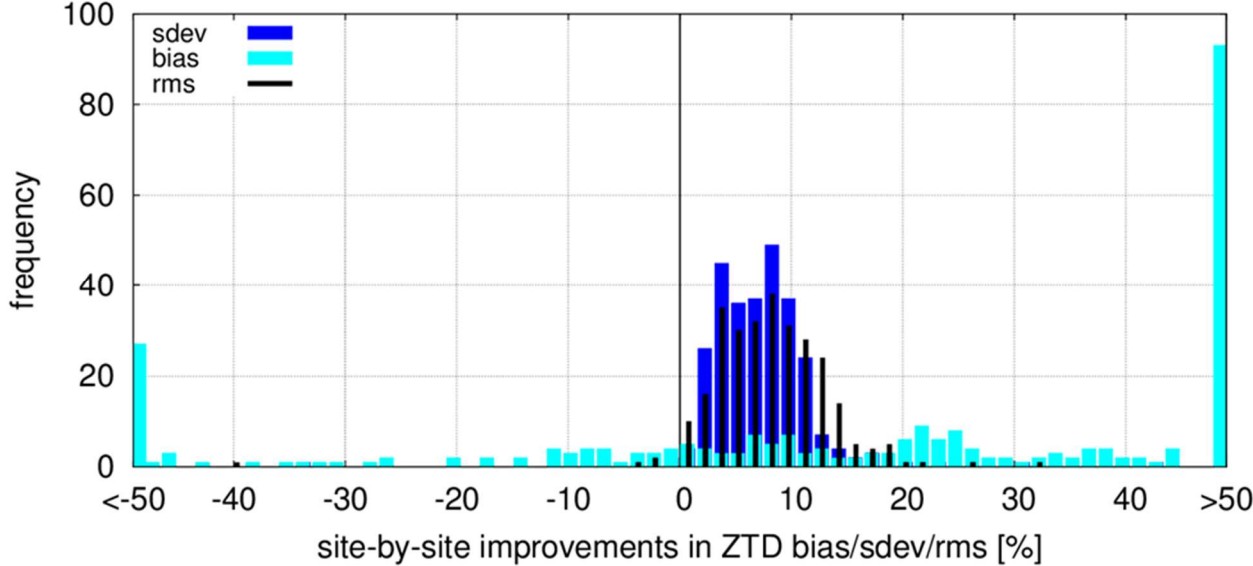

Figure 12: Site-by-site ZTD improvements of EPN-Repro2 versus EPN-Repro1 compared to ERA-Interim.

876

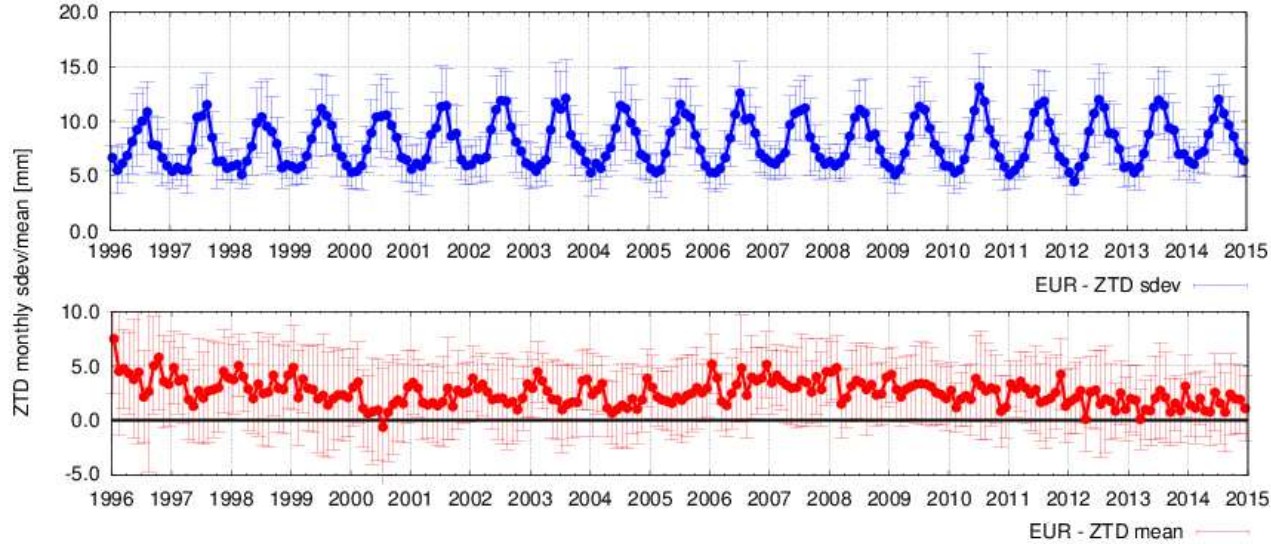

879

Figure 13: Time series of monthly mean biases (lower part) and standard deviations (upper part) for ZTD differences between EPN-Repro2 and ERA-Interim re-analysis (ERA-Interim minus GNSS). Uncertainties are calculated over all stations.

883

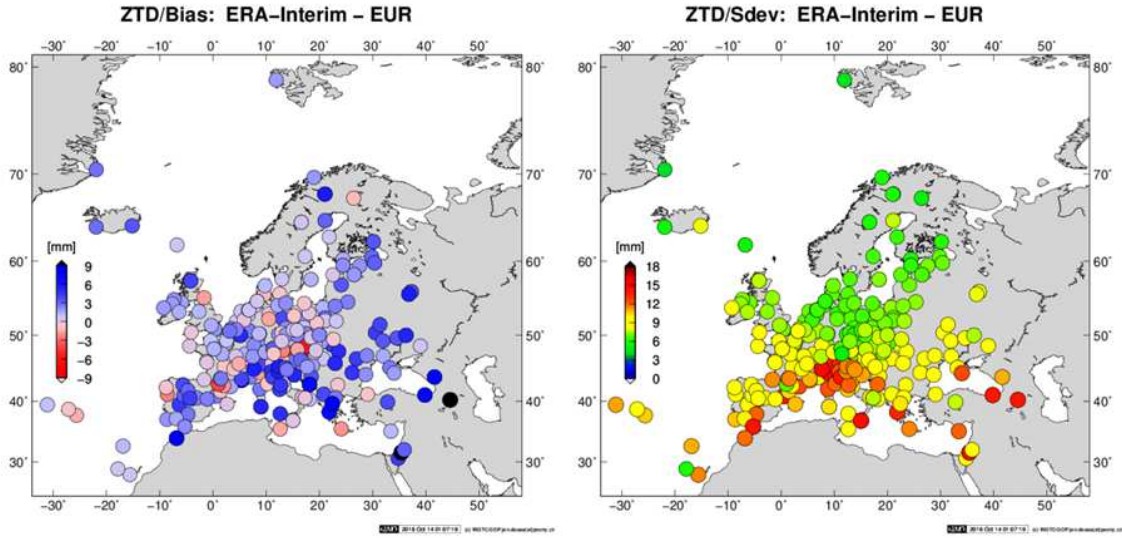

884

Figure 14: Geographical distribution of ZTD biases (left) and standard deviations (right) for EPN-
Repro2 compared to ERA-Interim (ERA-Interim minus GNSS).

887

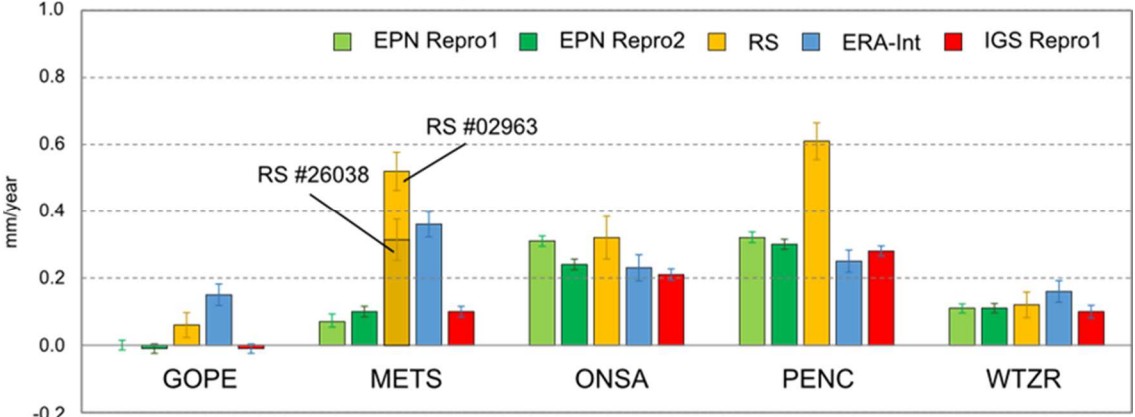

Figure 15: ZTD trend comparisons at five EPN stations for 5 different ZTD datasets. The error bars are the formal errors of the estimated trend values.