# Peer review of "EPN-Repro2: A reference GNSS tropospheric dataset over Europe."

_Atmospheric Measurement Techniques, 2016_

## Referee Comment (RC1) · Anonymous Referee #1 · 9 Jan 2017

As the GNSS tropospheric products are getting longer, it becomes more and more important to create homogenized products, especially for climate applications. From this perspective, the manuscript is timely and important. I think that the manuscript sill needs major revision before it is ready for publication. My main comments are two folds. First, I would like to see some more explanation on the differences (esp biases) presented from the comparisons with radiosonde and ERA-Int. There are a few specific comments listed below. Second, it would be great to show how the processed data improve the detection of PW trends, even with just a few examples. 1. Fig. 10: add a horizontal zero line, so that it would be easy to see the sign of the differences. This applies to other plots too. Any explanation to the statistically significant large biases? How does this compare with prior studies? It would be better to express the biases in percentage. 2. Fig. 11: I would recommend to add some quantitative numbers,

such as the reduction of biases and SDs, in the text (or Fig.) and the discussion. Based on visual examination, it looks like that it is mainly a shift 3. Fig.12, L357-358: It is not clear to me how "the limited temporal and horizontal NWM resolution as well as corresponding deficiencies in NWM orography" cause the negative differences in ZTD-NWM. Why does it vary with time (generally reduced magnitudes with time)?

---

## Referee Comment (RC2) · Anonymous Referee #2 · 16 Jan 2017

The article contributes to an important issue on homogenization and processing of GNSS tropospheric products for climate research. The article is timely and actual. It gives systematic overview about the reprocessing campaign and combination of data products from different ACs with additional attention on impact of GLONASS data, different antenna calibration models and non-tidal atmospheric loading. The results are evaluated with independent data sources (radiosondes and ERA-Interim) and illustrated with appropriate figures and tables. The article includes adequate references on related scientific research papers.

The manuscript needs some minor revision before getting ready for publication.

Figure 1: could look better with smaller markers.

The http-links should be checked. However, they may be broken only in this

version of discussion paper due to automatic document processing during its upload. In this case the remark on the next 3 links is not relevant. Line 36: https://www.iers.org/IERS/EN/Organization/AnalysisCoordinator/SinexFormat/sinex.html Line 45: http://www.euref.eu/documentation/MoU/EUREF-EUMETNET-MoU.pdf Line 566: http://www.epncb.oma.be/_documentation/papers/eurefsymposium2011/an_update_on

Line 230: small TYPO "... homogeneously reprocessed solutions (seeTable 2)".

Some questions:

Lines 346-352: Compared Repro1 and Repro2 with ERA-Interim, Figure 11, distribution of station means and standard deviations – over which time period the mean is calculated? ERA-Interim has 6 hrs time resolution, Repro1 and Repro2 have 1 hrs (Table 2). Could the result depend on interpolation made for synchronisation of timestamps for ERA-Interim and Repro2?

Lines 353-360: monthly mean biases, ZTD mean biases, Figure 12 – "There is no seasonal signal observed in time series of ZTD mean biases ...", but looking at the figure (upper part – monthly mean biases) – if it isn't a seasonal signal, then what is it?

---

## Author Response (AR1)

Response to Review #1.

**Overview**

As the GNSS tropospheric products are getting longer, it becomes more and more important to create homogenized products, especially for climate applications. From this perspective, the manuscript is timely and important. I think that the manuscript still needs major revision before it is ready for publication. My main comments are two folds. First, I would like to see some more explanation on the differences (esp biases) presented from the comparisons with radiosonde and ERA-Int. There are a few specific comments listed below. Second, it would be great to show how the processed data improve the detection of PW trends, even with just a few examples.

**Authors' Response**

The authors would like to thank Reviewer #1 for his/her constructive comments. We have considered them in the revised version to improve the quality of the paper.

We have reviewed section 4.1 'Evaluation versus Radiosonde', section 4.2 'Evaluation versus ERA-Interim' and section 5 'Conclusion' as reported below in 'Detailed Comments'.

**Detailed Comments**

**Reviewer # 1**

Fig. 10: add a horizontal zero line, so that it would be easy to see the sign of the differences. This applies to other plots too. Any explanation to the statistically significant large biases? How does this compare with prior studies? It would be better to express the biases in percentage.

**Authors' Response**

We have changed Figure 10 expressing the bias and standard deviation in percentage and adding the zero line as suggested. In Figure 10, we have modified the x-axis adding to the GPS site name the code of the Radiosonde used for the comparison. Moreover, we have compared the obtained results with prior studies available in literature and we have discussed mainly about the bias.

Below the revised version of section 4.1. 'Evaluation versus Radiosonde'. **Lines 286-297 changed:**

[revised manuscript text omitted]

**Reviewer # 1**

Fig. 11: I would recommend to add some quantitative numbers, such as the reduction of biases and SDs, in the text (or Fig.) and the discussion. Based on visual examination, it looks like that it is mainly a shift 3.

**Authors' Response**

The quantitative number for overall improvement from EUREF Repro1 to Repro2 was enumerated as 8-9 % for ZTD when considering total statistics in Table 4 while Figure 11 shows distributions of ZTD bias, standard deviation over all stations. Using data from Figure 11 we expressed site-by-site improvements of all statistics. Calculated median improvements for bias, standard deviation and RMS reached 21.1 %, 6.8 % and 8.0 %, respectively, which correspond with the value of 8-9 % for an overall improvement. An additional figure (not included in the revised text) shows the distribution of statistics of ZTD improvements over all stations. Degradation of standard deviation was found for three stations only, SKE8 (Skellefteaa, Sweden, integrated in the EPN since 28-09-2014), GARI (Porto Garibaldi, Italy, integrated in the EPN since 08-11-2009) and SNEC (Snezka, Czech Republic, former EPN station since 14-06-2009) all of them providing much less data compared to others, 1%, 30% and 3%, respectively. All other 290 stations showed improvements. We found 72 with increased absolute systematic errors in EUREF Repro1 compared to Repro2 while for all others (221 stations, 75%) systematic errors were reduced.

[Figure]

Below the revised version of section 4.2. 'Evaluation versus ERA-Interim'. **Lines 346-352 changed:**

> "For completeness, we evaluated also EPN Repro1 ZTD product with respect to the ERA-Interim using the same period, i.e. 1996-2014 when completed with the EUREF operational product after GPS week 1407 (December 30, 2006). Comparing Repro1 and Repro2 with the numerical weather re-analysis showed the 8-9% improvement of the latter in both overall standard deviation and systematic error. Figure 11 shows distributions of station means and standard deviations of EPN Repro1 and Repro2 ZTDs compared to NWM ZTDs using the whole period 1996-2014. Common reductions of both statistical characteristics are clearly visible for the majority of all stations. From data of the figure, we also expressed site-by-site improvements in terms of ZTD bias, standard deviation and RMS. Calculated medians reached 21.1 %, 6.8 % and 8.0 %, respectively, which corresponds to the abovementioned improvement of 8-9 %. The degradation of standard deviation was found at three stations: SKE8 (Skellefteaa, Sweden, integrated in the EPN since 28-09-2014), GARI (Porto Garibaldi, Italy, integrated in the EPN since 08-11-2009) and SNEC (Pod Snezkou, Czech Republic, former EPN station since 14-06-2009) all of them providing much less data compared to others, 1%, 30% and 3%, respectively. All other stations (290) showed improvements. We also found 72 stations with increased absolute bias in EUREF Repro1 compared to Repro2 while all others, 221 stations (75%), resulted in reduced systematic error."

**Reviewer # 1**

Fig.12, L357-358: It is not clear to me how "the limited temporal and horizontal NWM resolution as well as corresponding deficiencies in NWM orography" cause the negative differences in ZTD-NWM. Why does it vary with time (generally reduced magnitudes with time)?

**Authors' Response**

The corresponding sentence was finally removed. We have checked more individual stations at low altitude and the bias of -1 to 2 mm dominated even for those sites. Thus the dependence of an overall mean bias does not seem to be related to the limited spatial resolution or deficiencies in NWM orography as there was no observed significant difference in the Alps and within flat areas. The mean bias remains unknown and the uncertainty is still large and varying depending on a common set of stations.

**Reviewer # 1**

It would be great to show how the processed data improve the detection of PW trends, even just with a few examples.

**Authors' Response**

We have reviewed section 5 'Conclusion' and have added examples available in the literature. As an example of application of EPN Repro2 data, we cited, in addition to the assimilation trial ongoing at UK Met Office, comparisons with regional climate model simulations ongoing at Sofia University and Hungarian Meteorologic Service. **Lines 392-395 changed**:

"According to Wang et al. (2007) IGS ZTD products are valuable source of water vapor data for climate and weather studies. The GPS PW is useful also for monitoring the quality of the radiosonde data. However, a better spatial coverage of the GNSS PW data is needed to investigate and reduce systematic biases in comparison with the global radiosonde humidity data (Wang and Zhang, 2009). On the other hand extending the observation period and complement of temporal coverage is necessary to calculate more reliable mean values and trends. As it was pointed by Baldysz et al. (2015, 2016) additional two years of ZTD data can change estimated trends up to 10%. Therefore, data after 2010 and with a better coverage over

Europe are required for improving the knowledge of climatic trends of atmospheric water vapour in Europe. In this scenario, EPN-Repro2 can be used as a reference data set with a high potential for monitoring trend and variability in atmospheric water vapour. Comparisons with regional climate model simulations is one of the application of EPN-Repro2. Ongoing at Sofia University is comparison between GNSS IWV, computed from EPN-Repro2 ZTD data for SOFI (Sofia, Bulgaria), and ALADIN-Climate IWV simulations conducted by the Hungarian Meteorological Service, for the period 2003-2008. The preliminary results show a tendency of the model to underestimate IWV. Clearly, larger number of model grid points need to be investigated in different regions in Europe and the EPN-Repro2 data is well suited for this."

Baldysz, Z., Nykiel, G., Figurski, M., Szafranek, K., and Kroszczynski, K.: Investigation of the 16-year and 18-year ZTD Time Series Derived from GPS Data Processing. Acta Geophys. 63, 1103-1125, DOI: 10.1515/acgeo-2015-0033, 2015

Baldysz Z., Nykiel G., Araszkiewicz A., Figurski M. and Szafranek K.: Comparison of GPS tropospheric delays derived from two consecutive EPN reprocessing campaigns from the point of view of climate monitoring. Atmos. Meas. Tech., 9, 4861-4877, DOI: 10.5194/amt-9-4861-2016, 2016

Fig. 11: I would recommend to add some quantitative numbers, such as the reduction of biases and SDs, in the text (or Fig.) and the discussion. Based on visual examination, it looks like that it is mainly a shift 3.

**Authors' Response**

Taking into account what reported in the first response, we have decided to add the additional figure reported below. In the revised text it is Figure 12.

[Figure]

**Reviewer # 1**

It would be great to show how the processed data improve the detection of PW trends, even just with a few examples.

**Authors' Response**

We have reviewed section 5 'Conclusion' and have added examples available in the literature. As an example of application of EPN Repro2 data, we cited, in addition to the assimilation trial ongoing at UK Met Office, comparisons with regional climate model simulations ongoing at Sofia University and Hungarian Meteorologic Service.

As requested, we have computed ZTD trends at five EPN stations: GOPE (Ondrejov, Czech Republic, integrated in the EPN since 31-12-1995), METS (Kirkkonummi, Finland, integrated in the EPN since 31-12-1995), ONSA (Onsala, Sweden, integrated in the EPN since 31-12-1995), PENC (Penc, Hungary, integrated in the EPN since 03-03-1996) and WTZR (Bad Koetzting, Germany, integrated in the EPN since 31-12-1995) using EPN Repro2, EPN Repro1 completed with the EUREF operational products, radiosonde and ERA-Interim data. All of them are also in the IGS Network, for which IGS Repro1 time series is available. IGS Repro1 data completed with the IGS operational products have been extracted from the GOP-TropDB.

We have screened all data sets (classical 3 sigma). Then for all GPS ZTD data sets (EPN Repro2, EPN Repro1 + operational and IGS Repro1 + operational) we have estimated and removed shift related to the antenna replacement. No homogenization has been done for radiosonde since we do not have radiosonde metadata to do this properly. However, we think that this will affect the comparison of ZTD trends in the same way. A LSE method is applied to estimate trends and seasonal component.

Finally, we received trends for EPN Repro2 (GOPE=-0.01+/-0.014 mm/year; METS=0.10+/-0.016 mm/year; ONSA=0.24+/-0.016 mm/year; PENC=0.30+/-0.015 mm/year; WTZR=0.11+/-0.014 mm/year) and other data sets.

ZTD trends for all three GPS ZTD data sets are consistent, as soon as the same homogenisation procedure is applied. The overall RMS is 0.02 mm/year. Among all five ZTD sourced, we find the best agreement for ONSA (RMS=0.04mm/year) and WTZR (RMS=0.02mm/year). For PENC we have good agreement with respect to ERA-Interim (0.05 mm/year), but a large discrepancy versus radiosonde (-0.31 mm/year). This large discrepancy is probably due to the distance to the radiosonde launch site (40.7 km, radiosonde code 12843) and to the lack of the homogenisation stage. Over the five considered stations the agreement with respect to ERA-Interim (RMS = 0.11 mm/year) is better than that with respect to radiosonde (RMS = 0.16 mm/year). An additional figure (included in the revised text as Figure 15) shows the ZTD trend comparisons, the error bars are the formal error of the trend values.

[Figure]

For the considered stations EPN Repro2 do not change significantly the detection of ZTD trends as compared to EPN Repro1 + operational or IGS Repro1 + operational. However, it has generally a better agreement w.r.t. radiosonde and ERA-Interim data than EPN Repro 1 + operational. It has also the best spatial resolution than IGS Repro1 and radiosonde data, which are used today for long-term analysis over Europe. Taking into account the good consistency among trends, EPN Repro2 can be used for trend detection in areas where other data are not available.

**Lines 391-395 changed**:

"However, this data set is quite sparse over Europe (only 85 stations over the 280 EPN stations) and covers the period 1996-2010. According to Wang et al. (2007) IGS ZTD products are valuable source of water vapor data for climate and weather studies. The GPS PW is useful also for monitoring the quality of the radiosonde data. However, a better spatial coverage of the GNSS PW data is needed to investigate and reduce systematic biases in comparison with the global radiosonde humidity data (Wang and Zhang, 2009). On the other hand extending the observation period and complement of temporal coverage is necessary to calculate more reliable mean values and trends. As it was pointed by Baldysz et al. (2015,

2016) additional two years of ZTD data can change estimated trends up to 10%. Therefore, data after 2010 and with a better coverage over Europe are required for improving the knowledge of climatic trends of atmospheric water vapour in Europe. In this scenario, EPN-Repro2 can be used as a reference data set with a high potential for monitoring trend and variability in atmospheric water vapour. Considering five EPN stations, among those with the longest time span, GOPE (Ondrejov, Czech Republic, integrated in the EPN since 31-12-1995), METS (Kirkkonummi, Finland, integrated in the EPN since 31-12-1995), ONSA (Onsala, Sweden, integrated in the EPN since 31-12-1995), PENC (Penc, Hungary, integrated in the EPN since 03-03-2096) and WTZR (Bad Koetzting, Germany, integrated in the EPN since 31-12-1995), we have computed ZTD trends using EPN Repro2, EPN Repro1 completed with the EUREF operational products, radiosonde and ERA-Interim data. All of them are also in the IGS Network, for which IGS Repro1 completed with the IGS operational products are available and extracted from the GOP-TropDB. First we have removed annual signal from the original time series and marked all outliers according to 3-sigma criteria. Then for all GPS ZTD data sets we have estimated all well-known and recognized shifts related to the antenna replacement. No other unexplained breaks has been removed to be sure that we not introduce any artificial errors. Based on the cleaned and filtered data we have used linear regression model before and after the considered epoch independently. The difference between those two models in specific epoch is considered as a shift. Then, we have removed all the estimated shifts from the original time series. Generally, the size of the shifts is much lower than noise level and depends on the applied method of its estimation. Therefore, the final results are affected by used methodology and cannot be considered as an absolute values. No homogenization has been done for radiosonde since radiosonde metadata are not available. Finally, a LSE method have been applied to estimate linear trends and seasonal component. ZTD trends (Figure 14) for all three GPS ZTD data sets are consistent, as soon as the same homogenisation procedure is applied. Then overall RMS is 0.02 mm/year. Among all five ZTD sourced, we find the best agreement for ONSA (RMS=0.04mm/year) and WTZR (RMS=0.02mm/year). For PENC we have good agreement with respect to ERA-Interim (0.05 mm/year), but a large discrepancy versus radiosonde (-0.31 mm/year). This large discrepancy is probably due to the distance to the radiosonde launch site (40.7 km, radiosonde code 12843) and to the lack of the homogenisation stage. Over the five considered stations the agreement with respect to ERA-Interim (RMS = 0.11 mm/year) is better than that with respect to radiosonde (RMS = 0.16 mm/year). Even though for the five considered stations EPN Repro2 do not change significantly the detection of ZTD trends, it has a better agreement with respect to radiosonde and ERA-Interim data than EPN Repro1.It has also the best spatial resolution than IGS Repro1 and radiosonde data, which are used today for long-term analysis over Europe. Taking into account the good consistency among trends, EPN Repro2 can be used for trend detection in areas where other data are not available.

Comparisons with regional climate model simulations is one of the application of EPN-Repro2. Ongoing at Sofia University is comparison between GNSS IWV, computed from EPN-Repro2 ZTD data for SOFI (Sofia, Bulgaria), and ALADIN-Climate IWV simulations conducted by the Hungarian Meteorological Service, for the period 2003-2008. The preliminary results show a tendency of the model to underestimate IWV. Clearly, larger number of model grid points need to be investigated in different regions in Europe and the EPN-Repro2 data is well suited for this."

Baldysz, Z., Nykiel, G., Figurski, M., Szafranek, K., and Kroszczynski, K.: Investigation of the 16-year and 18-year ZTD Time Series Derived from GPS Data Processing. Acta Geophys. 63, 1103-1125, DOI: 10.1515/acgeo-2015-0033, 2015

Baldysz Z., Nykiel G., Araszkiewicz A., Figurski M. and Szafranek K.: Comparison of GPS tropospheric delays derived from two consecutive EPN reprocessing campaigns from the point of view of climate monitoring. Atmos. Meas. Tech., 9, 4861-4877, DOI: 10.5194/amt-9-4861-2016, 2016

The article contributes to an important issue on homogenization and processing of GNSS tropospheric products for climate research. The article is timely and actual. It gives systematic overview about the reprocessing campaign and combination of data products from different ACs with additional attention on impact of GLONASS data, different antenna calibration models and non-tidal atmospheric loading. The results are evaluated with independent data sources (radiosondes and ERA-Interim) and illustrated with appropriate figures and tables. The article includes adequate references on related scientific research papers. The manuscript needs some minor revision before getting ready for publication.

**Authors' Response**

The authors would like to thank Reviewer#2 for his/her constructive comments. We have considered them in the revised version to improve the quality of the paper.

**Detailed Comments**

**Reviewer # 2**

Figure 1: could look better with smaller markers.

**Authors' Response**

We have improved Figure 1 as below:

[Figure]

Figure 2. Time series of the number of GNSS observations for the period 1996-2014. GPS observations are shown in red, GPS+GLONASS in blue and their differences in green. The difference is significant starting 2008.

**Reviewer # 2**

The http-links should be checked. However, they may be broken only in this version of discussion paper due to automatic document processing during its upload. In this case the remark on the next 3 links is not relevant.

Line 36: https://www.iers.org/IERS/EN/Organization/AnalysisCoordinator/SinexFormat/sinex.html

Line 45: http://www.euref.eu/documentation/MoU/EUREF-EUMETNET-MoU.pdf

Line566:http://www.epncb.oma.be/_documentation/papers/eurefsymposium2011/an_update_on_epn_re processing_project_current_achievement_and_status

**Authors' Response**

Thank you for pointing this. We will check http-links in the final version of the manuscript.

**Reviewer # 2**

Line 230: small TYPO ": : : homogeneously reprocessed solutions (seeTable 2)".

**Authors' Response**

Correct.

**Reviewer # 2**

Lines 346-352: Compared Repro1 and Repro2 with ERA-Interim, Figure 11, distribution of station means and standard deviations – over which time period the mean is calculated? ERA-Interim has 6 hrs time resolution, Repro1 and Repro2 have 1 hrs (Table 2). Could the result depend on interpolation made for synchronisation of timestamps for ERA-Interim and Repro2?

**Authors' Response**

The mean in Figure 11 is computed for the period 1996-2014. The Repro1 dataset was completed with EUREF operational products after GPS week 1406 (December 23, 2006). For the comparison versus ERA-Interim we extracted 4 values per day at 00, 06, 12, 18 from Repro1 and Repro2 GNSS datasets using the linear approximation from values -/+ 30 min as EUREF solutions stores ZTDs in HR:30 only.

Datasets with different time resolutions affect the final comparison. However, both GNSS ZTD datasets (Repro1 and Repro2) have the same time resolution (1 hour) of values expressed at HR:30 and the interpolation affects both in the same way. Therefore, we can assume that both comparisons are compatible and the inter-comparison reflects principally the quality of products.

The following descriptions were improved in the manuscript:

**Lines 305-312 changed:**

"… software (Zus et al., 2014). Combined EUREF Repro1 and Repro2 products as well as individual ACs tropospheric parameters were assessed with the corresponding parameters estimated from the NWM re-analysis. The comparisons was done for the period 1996-2014 using the GOP-TropDB (Gyori and Dousa, 2016) via calculating parameter differences for pairs of stations and using values at every 6 hours (00:00, 6:00, 12:00 and 18:00) as available from the NWM product. A linear interpolation from values -/+ 30 min was thus necessarily applied for all GNSS products providing HH:30 timestamps as required for the combination process. As all compared GNSS products has the same time resolution (1 hour), the interpolation is assumed to affect all products in the same way. Therefore, we assume all inter-comparisons to a common reference (NWM) principally reflects the quality of the products. No vertical corrections were applied since NWM parameters were estimated for the long-term antenna reference position of each station."

**Lines 346-352 changed:**

"For completeness, we evaluated also EPN Repro1 ZTD product with respect to the ERA-Interim using the same period, i.e. 1996-2014 when completed with the EUREF operational product after GPS week 1407 (December 30, 2006). Comparing Repro1 and Repro2 with the numerical weather re-analysis showed the 8-9% improvement of the latter in both overall standard deviation and systematic error. Figure 11 shows distributions of station means and standard deviations of EPN Repro1 and Repro2 ZTDs compared to NWM ZTDs using the whole period 1996-2014. Common reductions of both statistical characteristics are clearly visible for the majority of all stations."

**Reviewer # 2**

Lines 353-360: monthly mean biases, ZTD mean biases, Figure 12 – "There is no seasonal signal observed in time series of ZTD mean biases …..:", but looking at the figure (upper part – monthly mean biases) – if it isn't a seasonal signal, then what is it?

**Authors' Response**

In Figure 12: description of the subplots is swapped. The caption of Figure 12 is correct as follows: "
[revised manuscript text omitted]

---

## Editor Decision (ED1)

1 EPN Repro2 EPN-Repro2: A reference GNSS tropospheric dataset over Europe.

[revised manuscript text omitted]

72 PromotingTo promote the use of reprocessed long-term GNSS-based tropospheric delay data sets for climate research is one of the objectives of the Working Group 3 'GNSS for climate monitoring' 73 74 of the EU COST Action ES 1206 'Advanced Global Navigation Satellite Systems tropospheric 75 products for monitoring severe weather events and climate (GNSS4SWEC)', launched for the 76 period of 2013-2017. The Working Group 3 enforces the cooperation between geodesists and 77 climatologists in order to generate recommendations on optimal GNSS reprocessing algorithms for 78 climate applications, and to standardise for these applications the conversion method of conversion 79 between propagation delay and atmospheric water vapour,-(Saastamoinen, (1973);, Bevis et al., (1992); Bock et al. (2015), with respect to climate standards. For climate applications, maintaining 80 the long-term stability is a key issue. Steigenberger et al. (2007) found that the lack of consistencies 81 over time due to changes in GNSS processing could cause inconsistencies of several millimetres in 82 83 the GNSS-derived Integrated Water Vapour (IWV), making climate trend analysis very challenging. Jin et al. (2007) studied the seasonal variability of GPS Zenith Tropospheric Delay (1994-2006) 84 over 150 international GPS stations and showed itsthe relative trend in the northern hemisphere and 85 southern hemisphere as well as in coastal and inland areas. Wang and Zhang (2009) derived GPS 86 Precipitable Water Vapour (PWV or PW) using the International GNSS Service (IGS,), Dow et al., 87 (2009), tropospheric products at about 400 global sites for the period 1997-2006 and analysed the 88 PWV diurnal variations. Nilsson and Elgered (2008) showed reported on PWV changes from -0.2 89 mm to +1.0 mm in 10 years by using the data from 33 GPS stations located in Finland and Sweden. 90 Sohn and Cho (2010) analysed the GPS Precipitable Water Vapour trend in South Korea for the 91 period 2000-2009 and examined studied also the relationship between GPS PWV and temperature, 92 93 which is the one of the climatic elements. Better information about A more thorough knowledge of atmospheric humidity, particularly in climate-sensitive regions, is essential to improve the diagnosis 94 of global warming, and for the validation of climate predictions on which socio-economic response 95 strategies-are based with strong societal benefits. Suparta (2012) reported onpointed out that the 96 validation of PWV ais an essential tool for solar-climate studies over a tropical region. Ning et al. 97 (2013) used 14 years of GPS-derived IWV at 99 European sites to evaluate the regional Rossby 98 Centre Atmospheric (RCA) climate model. GPS monthly mean data were compared against RCA 99 simulations and the-ERA\_-Interim data. Averaged over the domain and the 14 years covered by the 100 GPS data, they found IWV differences of about 0.47 kg/m2 and 0.39 kg/m2 for RCA-GPS and 101

102 ERA-interimCMWF-GPS, with a standard deviations of 0.98 kg/m2 and whereas it is 0.35 kg/m2, 
[revised manuscript text omitted]

**Comment [g7]:** Add a bookmark here explaining that TRM55971.00 points to the provider and the type of the antenna. Simila for TZGD radome.

Comment [g8]: Another TZGD radome?

**Comment [g9]:** For all of them: do you me antennas AND radomes??? Please be more specific.

component (vertical displacement) is  $-5.2 \pm 0.5$  mm,  $8.7 \pm 0.6$  mm and  $5.6 \pm 0.8$  mm with a 201 202 corresponding offset in the ZTD of  $0.2 \pm 0.5$  mm,  $-1.5 \pm 0.5$  mm,  $-1.4 \pm 0.8$  mm, respectively. Similar situation appears also values were obtained between solutions calculated for all 203 204 stations/antennas for which individual calibration models are available. The corresponding offset in the ZTD has the opposite sign for the antennas with an offset in the up component larger than 5 mm 205 (16 antennas) and, generally, does not exceeding 2 mm for ZTD. Such inconsistencies in the ZTD 206 time series are not large enough to be captured during the combination process (see Section 3), 207 where a 10 mm threshold in the ZTD bias (about  $1.5 \text{ kg/m}^2$  IWV) is set in order to flag problematic 208 ACs or stations. 209

**210 2.3 Impact of non-tidal atmospheric loading**

As reported in the IERS Convention (2010), the diurnal heating of the atmosphere causes surface 211 pressure oscillations at diurnal S1, semidiurnal S2, and higher harmonics. These atmospheric tides 212 induce periodic motions of the Earth's surface (Petrov and Boy, 2004). The conventional 213 recommendation is to calculate the station displacement using the Ray and Ponte (2003) S2 and S1 214 215 tidal model. However, crustal motion related to non-tidal atmospheric loading has been detected in 216 station position time series from space geodetic techniques (van Dam et al., 1994; Magiarotti et al., 217 2001, Tregoning and Van Dam, 2005). Several models of station displacements related to this effect 218 are currently available. Non-tidal atmospheric loading models are not yet considered as Class-1 models by the International Earth Rotation and Reference Systems Service (IERS 2010), indicating 219 220 that there are currently no standard recommendations for data reduction. To evaluate their impact, 221 two solutions, one without and one without a non-tidal atmospheric loading model, have been 222 compared for the year 2013. In the last one the solution with the model, the National Centers for 223 Environmental Prediction (NCEP) model is used at the observation level during data reduction 224 (Tregoning and Watson, 2009).

225 Dach et al. (2010) have already found that the repeatability of the station coordinates improves by 226 20% when applying the non-tidal atmospheric loading correction effect-directly on the data analysis 227 and by 10% when applying a post-processing correction to the resulting weekly coordinates compared with a solution without considering these corrections. However, the effect of applying 228 non tidal atmospheric loading on the ZTDs seems to be negligible. Generally, it causes a difference 229 230 below 0.5 mm with a scattering standard deviation not larger than 0.3 mm. The difference is thus below the level of confidence. Figure 4Figure 4 shows time series of the differences of the ZTDs 231 232 and the up components between two tsolutionsime series obtained with and without non-tidal atmospheric loading for two EPN stations: KIR0 (Kiruna, Sweden) and RIGA (Riga, Latvia). 233

**Comment [g10]:** What does the S1 and S2 stand for? Could you just say "surface press oscillations with diurnal, semidiurnal variabi and even higher harmonics?"

**Comment [g11]:** Same comment as here above.

Furthermore, tThere is also no correlation between the values of estimated differences and vertical
 displacements caused by non-tidal atmospheric loading, as- cCorrelation coefficients for the
 analysed EPN stations were below 0.2.

**237 **3.** EPN--Repro2 combined solutions**

[revised manuscript text omitted]

**Comment [g14]:** On which grounds? Plea comment!

Black, Check spelling and grammar

**Comment [g15]:** Height component = up component,? If not, please explain the difference. Please use a consistent term (up component or height component) in order t not mislead less GNSS experienced readers.

**Comment [g16]:** So, it turns out that the threshold is 9 mm in the up comment (see a Fig. 7). So, why are so speaking about this 16mm differences in the up component? Th is totally not clear to me. Please explain.

**300 4. Evaluation of the ZTD Combined Products with respect to independent data sets**

The evaluation with respect to other sources or products, such as rRadiosonde data from the E-GVAP and numerical weather re-analysis from the European Centre for Medium-Range Weather Forecasts, ECMWF (ERA-Interim), provides a measure of the accuracy of the ZTD combined products.

305 4.1 Evaluation versus radiosonde

For the GPS and rRadiosonde (RS) comparisons at the EPN collocated sites, we used profiles from 306 the World Meteorological Organization (WMO) provided by EUMETNET in the framework of the 307 Memorandum of Understanding between EUREF and EUMETNET. Radiosonde profiles are 308 309 processed using athe\_software by (Haase et al., (2003) that checks the quality of the profiles, converts the dew point temperatures to specific humiditiesy, shiftstransforms the radiosonde profile 310 311 to correct for the altitude offset between the GPS and the radiosonde sites, and determines the ZTD, Zenit Wet Delay and IWV compensating for the change of the gravitational acceleration,  $g_{\tau}$  with 312 height. 313

314 A comparision of the GNSS and radiosonde ZTD time series for the EPN site CAGL (Cagliari, 315 Sardinia Island, Italy) is shown in Figure 9<del>Figure 9</del>, with the mean biases and standard deviations reported in the Figure. -shows an example for the EPN site CAGL (Cagliari, Sardinia Island, Italy). 316 Similarly, For all the 183 EPN collocated sites, and using all the data available in the considered 317 period, we computed an overall bias (RS minus GNSS) and standard deviation for all the 183 EPN 318 319 collocated sites, using all the data available in the considered period (Figure 10Figure 10). In this figure, tThe sites are sorted withaccording to the increasing distances from the nearest rRadiosonde 320 321 launch site. For instance, MALL (Palma de Mallorca, Spain) is the closest (0.5 km to the rRadiosonde site with WMO code code 8301) while GRAZ (Graz, Austria) is the most distant (133 322 323 km to\_Radiosonde code RS WMO code 14015). The amount of data available for the comparisons varies between sites, depending on the availability of the GPS and rRadiosonde ZTD estimates in 324 the considered epoch, and it-ranges from 121 pairs for VIS6 (Visby, Sweden, integrated in the EPN 325 since 22-06-2014) up to 21226 pairs for GOPE (Ondrejov, Czech Republic, integrated in the EPN 326 327 since 31-12-1995).

The mean bias ranges from  $-0_{17}87\%$ , which corresponds to  $-21_{17}2$  mm in ZTD7 (at EVPA, Ukraine, atnd a distance of 96.5 km from the RS WMO-Radiosonde launch site 96.5 km, Radiosonde code 330 33946 station) to  $0_{17}68\%$ , which corresponds to  $15_{17}4$  mm7 (at OBER, Germany at, and distance from the Radiosonde launch site 90.8 km from RS WMO, Radiosonde code 11120). The overall mean ZTD bias for all sites is  $-0_{17}6$  mm with a standard deviation of 4.9 mm. For the more than **Comment [g17]:** The ZWD is for the first time used here. Please explain how it is defined or calculated. If not, just do not mention it.

**Comment [g18]:** In Fig. 9, the bias is define at RS ZTD minus GNSS ZTD. I assume that you also used this definition of the bias through the entire paper?

Formatted: Font: Not Italic, Font color: Black, English (U.K.) 333 75% of the stations (178 pairs), the agreement is below 5 mm in ZTD and only 5.5% of the stations (13 pairs) have ZTD biases higher than 10 mm. The higher biases concernarise mostly for the 334 paired sitess over 50 km away from each other, for which differences in the geographical 335 336 representativeness become important. For example, the, like GPS stations OBER, OBE2 and OBET located in Oberpfaffenhofen (Germany) areand collocated with the RS WMO - Radiosonde 337 (VRS90L code 11120) at launched from Innsbruck Airport in Austria, on the opposite side of the 338 North Chain in the Karwendel Alps. Our results are at odds with Wang et al. (2007), in which 339 where the authors compared PW (not ZTD) from GPS and global rRadiosondes, in the sense that .- In 340 contrast to them, we found received a small negative bias -1.19 mm for Vaisala rRadiosondes, which 341 is the most common type used in Europe (81% of all used in this study). It should be however noted 342 that different Vaisala radiosonde types (e.g. RS80 vs RS90/RS92) are equipped with different 343 humidity sensors, resulting in e.g. different RS-GPS comparisons in PW (e.g. Van Malderen et al., 344 2014 and references therein). For MRZ, GRAW and M2K2 readiation represent 345 4.6%, 3.4% and 3.0% of the compared rRadiosondes types respectively, we received a systematic 346 positive bias. However, Wang et al. (2007) used global Radiosonde data from 2003 and 2004, while 347 we used all available data over Europe from 1994 to 2015. This can partly explain the disagreement 348 even though more analysis deserves to be done. Further investigation is also needed for several near 349 or moved GPS stations. For example in Brussels (Belgium) BRUS station, included in the EPN 350 351 network since 1996, was replaced by BRUX in 2012. Their bias w.r.t. the same Radiosonde (WMOVRS80L code 6447) has opposite sign (-1.2 mm and 3.4 mm respectively). A possible 352 353 explanation is the different time span over which the bias has been computed (1996-2012 for BRUS, 2012-2015 for BRUX). 354

In agreement with Ning et al. (2012), the ZTD standard deviation generally increases with the 355 distance from the rRadiosonde launch site. It is in the range of  $[0_{27}16; 0_{27}76]$  %, which corresponds 356 to [3; 18] mm in ZTD, till 15 km (first band in Figure 10); in [0.729; 0.78] % , which 357 correspondings to [7; 19] mm, till 70 km (second band in Figure 10), and in [10; 33] mm till 133 358 359 km (third band in Figure 10). The evaluation numbers of the standard deviation areis comparable with previous studies. Haase et al. (2001) showed a very good agreement with biases less than 5 360 mm in ZTD and athe standard deviation of 12 mm for most of the analysed sites in Mediterranean. 361 362 Similar results (6.0 mm  $\pm$  11.7 mm) were obtained also by Vedel et al. (2001). Both of themstudies 363 were based on non-collocated pairs at sites distant-less than 50 km from each other. Pacione et al 364 (2011), considering 1-year of GPS ZTD and rRadiosonde data over the E-GVAP super sites 365 network, obtained a standard deviation of 5-14 mm. Dousa et al. 2012 evaluated ZTDs from GNSS

**Comment [g19]:** This is the radiosonde ty Vaisala RS90.

**Comment [g20]:** M2K2 is a radiosonde ty that Olivier Bock has assessed in his AMT pa (doi:10.5194/amt-6-2777-2013). Please comment if in this paper also a positive bias has been found.

**Comment [g21]:** Wang et al. (2007) report about a dry bias in PW of the radiosondes werespect to the GNSS retrievals of 1.08 mm (drier in the radiosondes). If the information Fig. 9 is correct and your bias is also calculat as RS – GPS, you also find a negative (hence bias for the radiosondes, compared to GNSS So the results here agree with the Wang et a (2007) results. By the way, a description of the origin and references of the dry biases in Vaisala radiosondes can also be found in ou AMT 2014 paper (doi:10.5194/amt-7-2487-2014). The reference document is https://www.wmo.int/pages/prog/www/IW /publications/IOM-107\_Yangjiang.pdf

**Comment [g22]:** VRS80L is the radiosond type: Vaisala RS80.

**Comment [g23]:** In our AMT2014 paper, section 4.2, we also make the RS-GPS comparison separately for the two (or 3) radiosonde types that have been used at Brussels. As a matter of fact, we switched fri RS80 to RS90 in August 2007. You are hence comparing BRUS with RS80 and RS90/RS92, while comparing BRUX only with RS92. This have a non-neglible effect on the compariso and is hence not related only to the change the GPS station!

and Rradiosondes on a global scale over a 10-month period and reported a standard deviation of 5-366 367 16 mm.

If we compare both the EPN-Repro1 ZTD product (completed with the EUREF operational product 368 after 30 December 2006) and the EPN-Repro2 with the radiosonde ZTDs for the same period 1996-369 370 2014, we found an improvement of approximately 3-4% in the overall standard deviation for the 371 second processing. The assessment of the EPN Repro1-EPN-Repro1-ZTD product with respect to 372 Radiosonde using the same period, i.e. 1996 2014 when completed with the EUREF operational 373 product after GPS week 1407 (December 30, 2006), and EPN Repro2 with respect to the Radiosonde data has an improvement of approximately 3-4% in the overall standard deviation. 374 375 4.2 **Evaluation versus ERA-Interim data** We also compared the EPN-Repro2 ZTDs with the ZTDs calculed from ERA-Interim (Dee et al., 376 377 2011) from the European Centre for Medium-Range Weather Forecasts (ECMWF)) are used as Numerical Weather Prediction (NWP) model data. The ERA-Interim is a re-analysis product of a 378 379 Numerical Weather Prediction (NWP) model and is available every 6 hours (00, 06, 12, 18 UTC)

380 with a horizontal resolution of  $1 \times 1$  degree and with 60 vertical model levels.

For the period 1996-2014 and for each EPN station, the ZTD and tropospheric linear horizontal 381 382 gradients were computed using the GFZ (German Research Centre for Geosciences) ray-tracing software (Zus et al., 2014). Combined EUREF Repro1 and Repro2 products as well as individual 383 ACs tropospheric parameters were assessed with the corresponding parameters estimated from the 384 NWMERA-interim re-analysis. The evaluation of GNSS and NWMERA-interim was performed 385 using the GOP-TropDB (Gyori and Dousa, 2016) byvia calculating parameter (ZTD, horizontal 386 gradients, see below) differences for each station pairs of stations, using the values at every 6 hours 387 (00:00, 06:00, 12:00 and 18:00), -as available from the NWMERA-interim model outputproduct. A 388 389 linear temporal interpolation to those four timestamps from values /+ 30 min-was thus necessarily applied for all GNSS products, which are available in-providing HH:30 timestamps as required for 390 the combination process. As all compared GNSS products haves the same time resolution (1 hour), 391 the interpolation is assumed to affect all products in the same way. Therefore, we assume that all 392 inter-comparisons to a common reference (NWMERA-interim) principally reflects the quality of 393 the products. No vertical corrections were applied since **NWMERA**-interim variablesparameters 394 395 were estimated for the long-term antenna reference position of each station.

Table 4 Table 4 summarizes the mean total statistics of individual (ACs) and combined (EUREF) 396 tropospheric parameters, ZTDs and horizontal gradients, over all available stations. The EUREF 397 398

combined solution does not provide tropospheric gradients and these could therefore be evaluated

399 for individual solutions only. In Table 4<del>Table 4, we can observe</del> a common ZTD bias (GNSS minus 400 ERA-interim, of about -1.8 mm is found for all GNSS solutions compared to the ERA-Interim, however still highly varying for individual stations as obvious from estimated uncertainties but a 401 402 large station to station variability could be noted, as is obvious from the estimated uncertainties. ZTD standard deviations are generally at the level of 8 mm between GNSS and NWMERA-interim 403 404 ZTDsproducts, but with thefor IGO solution performing about 25% worse than the others as already detected during the combination. Two solutions, AS0 and LP1 are slightly better than GO4 and 405 MU2: with a reaching the standard deviation of 7.7 mm, their accuracy is at the level of the 406 EUREF combined solution. The better performance of the ASO solution can be explained by 407 considered applying a stochastic troposphere modelling using undifference observations sensitive to 408 the absolute tropospheric delays, so that the due to its theoretical better capability of the modelling 409 true dynamics in the troposphere is better taken into account. as the solution applied a stochastic 410 troposphere modelling using undifference observations sensitive to the absolute tropospheric delays. 411 On the other hand, LP1 included roughly one third from of the EPN stations, which were properly 412 selected according to the station quality, herebythus making it difficult a difficulty to interpret thise 413 difference with respect to those solutions processing the full EPN. 414

415 The comparison of tropospheric linear horizontal gradients (East and North) from GNSS and NWMERA-interim revealed a problem with the MU2 solution (see Table 4). This solution 416 showsing a high inconsistency of results over different stations, which is not visible in the total 417 statistics, but mainly in the uncertainties, which are by an order or magnitude higher compared to 418 all other solutionss. A gGeographical plot (not shown hereed) confirmed this site-specific 419 420 systematic effect, but in both in positive and negative senses. The impact was however not observed in the MU2 ZTD results. Additionally, the GO4 solution performed slightly worse than the others. 421 ThisIt was identified as a consequence of estimating 6-hour gradients using athe piece-wise linear 422 function and without any absolute or relative constraints. In such case, higher correlations with 423 424 other parameters occurred and increased theraising uncertainties of the estimates. For this purpose, 425 the GO6 solution (not showned) was derived, fully compliant with the GO4, but stacking tropospheric gradients into 24 hours piece-wise linear modelling. In comparison with the former 426 427 GO4 solution By comparing the GO6 (Dousa and Vaclavovic, 2016), the GO6 standard deviations 428 dropped from 0.38 mm to 0.28 mm and from 0.40 mm to 0.29 mm for East and North gradients, 429 respectively, which corresponds to the LP1 solution that applied ying the same settings. Additionally, 430 Dousa and Vaclavovic (-2016) found a strong impact of a low-elevation receiver tracking problem 431 on the estimation of the horizontal gradients, which was particularly visible when comparing withed to the ERA-Interim horizontal gradients. Looking for ssystematic behaviour in monthly mean 432

**Comment [g24]:** Please specify how the b is calculated.

**Comment [g25]:** What do you mean by "undifference"?

differences in the gradients therefore seems to be a useful indicator for instrumentation-related
issues and should be applied as one of the tools for cleaning the EPN historical archive.

For completeness, we also evaluated thealso EPN--Repro1 ZTD product with respect to the-ERA-435 Interim using the same period, i.e. 1996-2014 (after-when- completing againged with the EUREF 436 437 operational product, see above) after GPS week 1407 (December 30, 2006). Comparing EPN Repro1EPN-Repro1 and EPN-Repro2EPN-Repro2 with the numerical weather model re-analysis 438 showed athe 8-9% improvement of EPN-Repro2the latter in both overall standard deviation and 439 biassystematic error. Figure 11Figure 11 shows the distributions of station means biases and 440 standard deviations of EPN-Repro1EPN-Repro1 and EPN-Repro2EPN-Repro2 ZTDs compared to 441 442 NWMERA-interim ZTDs using the whole period 1996-2014. Common reductions of both statistical characteristics are clearly visible for the majority of all stations. From the data of Figure 11Figure 443 444 11, we also illustrate the expressed site-by-site improvements in terms of ZTD bias, standard deviation and RMS in(Figure 12Figure 12). The cCalculated median improvements for these 445 446 statistics s-reached 21.1 %, 6.8 % and 8.0 %, respectively, which corresponds to the 447 abovementioned improvement of 8-9 %. AThe degradation of the standard deviation was found at 448 three stations: SKE8 (Skellefteaa, Sweden, integrated in the EPN since 28-09-2014), GARI (Porto Garibaldi, Italy, integrated in the EPN since 08-11-2009) and SNEC (Snezka, Czech Republic, 449 former EPN station since 14-06-2009). These 3 stations all of them provideing much less data 450 compared to other stations, 1%, 30% and 3%, respectively. All other stations (290) showed 451 improvements. We also found 72 stations with increased absolute bias in EPNUREF-Repro21 452 compared to Repro12 while athe other ll others, 221 stations (75%) had a - resulted in reduced bias 453 454 with ERA-interim ZTD.systematic error.

455 Time series of monthly mean biases and standard deviations for ZTD differences of EPN 456 Repro2EPN-Repro2 and the-ERA-Interim areis showned in Figure 13Figure 13. The small negative bias slowly decreases towards 2014, but thea high uncertainty of the mean bias indicates a site-457 458 specific behaviour, depending mainly on latitude and altitude of the EPN station and the quality of both **NWMERA-interim** and GNSS products. There is almost no seasonal signal observed in the 459 460 time series of ZTD mean biases or the uncertaintiesy, but clearly in the ZTD mean standard deviation and the uncertaintiesy. The sslightly increasing standard deviation towards 2014 can be 461 462 attributed to the increase of number of stations in EPN: starting from about 30 in 1996 and with more than 250 in 2014. A higher number of More stations reduces thea variability in monthly mean 463 464 biases, however, site-specific errors then contribute more to higher values of standard deviation.

**Comment [g26]:** 1 or 3% less data compa to the other stations is not very significant, So, what is the main reason that those statii have larger standard deviations for Repro2 ERA-interim versus Repro1 – ERA-interim?? Please explain.

**Comment [g27]:** I guess the order of Repu and Repro 1 should be changed in this sentence (otherwise EPN-Repro1 would be closer to ERA-interim than EPN-Repro2). Ple check!!!

**Comment [g28]:** Explain shortly where th seasonality comes from.

465 Figure 14Figure 14 displays the geographical distribution of total ZTD biases and standard 466 deviations for all sites. Prevailing negative biases seem to become lower or even positive in the 467 mountain areas. There is no latitudinal dependence observed for ZTD biases in Europe, but a strong 468 one for standard deviations. This corresponds mainly to the increase of water vapour content and its 469 variability towards the equator.

**470 **4.3 Evaluation of trends**

496

471 To illustrate the impact of the new processing on the resulting ZTD trends and uncertainties, we 472 considered five EPN stations, among those with the longest time span: GOPE (Ondrejov, Czech Republic, integrated in the EPN since 31-12-1995), METS (Kirkkonummi, Finland, integrated in 473 474 the EPN since 31-12-1995), ONSA (Onsala, Sweden, integrated in the EPN since 31-12-1995), PENC (Penc, Hungary, integrated in the EPN since 03-03-2096) and WTZR (Bad Koetzting, 475 Germany, integrated in the EPN since 31-12-1995). For these 5 stations, we have computed ZTD 476 trends using EPN-Repro2, EPN-Repro1 (again completed with the EUREF operational products), 477 radiosonde and ERA-Interim data. Furthermore, those 5 stations also belong to the IGS Network, 478 for which IGS Repro1, completed with the IGS operational products, are available and extracted 479 from the GOP-TropDB, so that we could also calculate ZTD trends from this dataset. 480 First, we removed the annual signal from the original time series and marked all outliers according 481 482 to the 3-sigma criterion. Then, we tried to remove all inhomogeneities in the GPS ZTD time series, related to instrumental changes, which might introduce a change in the mean of the ZTD time series 483 484 and therefore have an impact on the ZTD trends. In particular, for all GPS ZTD data sets we have estimated all documented shifts in the mean related to the antenna replacement. No other 485 unexplained break points has been corrected for, to be sure not to introduce any artificial errors. 486 Based on these cleaned and filtered data, we have used, independently, a linear regression model 487 before and after the considered epoch of the offset. The difference of the mean ZTDs between those 488 two linear regression models is then considered as the offset of the specific epoch is. With this 489 technique, we removed all the estimated offsets from the original GPS ZTD time series. Generally, 490 the amplitudes of the offsets are much lower than the noise level and depend on the applied method 491 of estimation. Therefore, the final ZTD trends and uncertainties presented here are affected by the 492 used methodology and should not be considered in absolute terms. No homogenization has been 493 done for the radiosonde data, since reliable metadata are not available. Also the ERA-interim ZTD 494 time series were not corrected for inhomogeneities. Finally, a Least Squares Estimation method has 495

been applied to estimate the linear trends and the seasonal components.

| 497 | In Figure 15Figure 15, the ZTD trends and uncertainties are presented for the 5 sites and for all  |
|-----|----------------------------------------------------------------------------------------------------|
| 498 | ZTD datasets. First of all, it should be noted that the trends between the three GPS ZTD data sets |
| 499 | are very consistent (as long as the same homogenisation procedure is applied). The overall RMS is  |
| 500 | 0.02 mm/year. If we now consider all five ZTD sources, the best agreement between the ZTD          |
| 501 | trends is achieved at ONSA (RMS=0.04mm/year) and WTZR (RMS=0.02mm/year). For PENC, we              |
| 502 | also have a good agreement of the GPS ZTD trends with respect to ERA-Interim (0.05 mm/year),       |
| 503 | but a large discrepancy with the radiosonde ZTD trend is found (-0.31 mm/year). This large         |
| 504 | discrepancy is probably due to the distance to the radiosonde launch site (40.7 km, RS WMO 12843)  |
| 505 | and to the lack of homogenization of the radiosonde data. For the five considered stations, the    |
| 506 | agreement of GPS ZTD trends with respect to ERA-Interim (RMS = 0.11 mm/year) is better than        |
| 507 | with respect to radiosondes (RMS = 0.16 mm/year). Even although, for the five considered stations, |
| 508 | EPN-Repro2 do not change significantly the value of the ZTD trends with respect to EPN-Repro1,     |
| 509 | it has a slightly better agreement with the radiosonde and ERA-Interim ZTD trends. Over Europe,    |
| 510 | the EPN network also has a better spatial resolution than the IGS and radiosonde networks, which   |
| 511 | are used today for an observations-based long-term analysis of ZTD/IWV variability over Europe.    |
| 512 | Taking into account the good consistency among the ZTD trends, EPN-Repro2 can be used for          |
| 513 | trend detection in areas where other data are not available.                                       |

**Comment [g29]:** Overall RMS between th GPS ZTD trends?

**Comment [g30]:** Add a number here to proof this statement.

**Comment [g31]:** I think this analysis deserves a separate secton and does not belong to the conclusions. In the conclusion you should just give a wrap up, and not describing new research. I also seriously changed the text. Please go through it and check thoroughly if you could life with every modification I propose!

**515 **5.** Conclusions**

514

516 In this paper, we described the activities carried out in the framework of the EPN second 517 reprocessing campaign. We focused on the tropospheric products homogenously reprocessed by 518 five EPN Analysis Centres for the period 1996-2014 and we described the ZTD combined products.

Both individual and combined tropospheric products, along with reference coordinates and other
metadata, are stored in a\_SINEX TRO format (,-Gendt, G. (1997), and are available to the users at
the EPN Regional Data Centres (RDC), located at BKG (Federal Agency for Cartography and
Geodesy, Germany). For each EPN station, plots on ZTD time series, ZTD monthly means,
comparison withversus Rradiosonde data (if collocated), and comparison versus the ERA-Interim
data will be available at the EPN Central Bureau (Royal Observatory of Belgium, Brussels,
Belgium).

We showed that EPN-Repro2 led to an improvement of approximately 3-4% in the overall standard
 deviation in the ZTD differences with radiosonde data, as compared with Assessment of the EPN
 Repro1EPN-Repro1. and Repro2 with respect to the Radiosonde data has an improvement of
 approximately 3-4% in the overall standard deviation.

530 The aAssessment of the EPN Repro1EPN-Repro21 and Repro2 with respect to comparison with the 531 ERA-Interim re-analysis showed athe 8-9% improvement of the latter over the former-in both the 532 overall ZTD bias and standard deviation with respect to EPN-Repro1 and systematic error which 533 was obvious for the majority of the stations. Comparisons of the GNSS solutions with the 534 NWMERA-interim, i.e. independent source, showed the overall agreement at the level of 8-9 mm, 535 however, rather site-specific ranging from 5 mm to 15 mm for standard deviations and from -7 mm 536 to 3 mm for biases considering 99% of results roughly.

537 The use of ground-based GNSS long-term data for climate research is an emerging field. For example, for the assessment of Euro-CORDEX (Coordinated Regional Climate Downscaling 538 539 Experiment) climate model simulation, the IGS Repro1dataset (-Byun and Bar-Sever, (2009), has 540 been used as reference reprocessed GPS products (Bastin et al. 2016). However, this data-set is 541 quite sparse over Europe (only 85 stations over the 280 EPN stations) and covers only the period 1996-2010. According to Wang et al. (2007) IGS ZTD products are valuable source of water vapor 542 data for climate and weather studies. The GPS PW is useful also for monitoring the quality of the 543 radiosonde data. However, a better spatial coverage of the GNSS PW data is needed to investigate 544 545 and reduce systematic biases in comparison with the global radiosonde humidity data (Wang and Zhang, 2009). On the other hand extending the observation period and complement of temporal 546 coverage is necessary to calculate more reliable mean values and trends. As it was pointed by 547 Baldysz et al. (2015, 2016) an additional two years of ZTD data can change the estimated trends up 548 to 10%. Therefore, with data after 2010 and with a better coverage over Europe, are required for 549 improving the knowledge of climatic trends of atmospheric water vapour in Europe. In this scenario, 550 551 EPN-Repro2 can be used as a reference data set with a high potential for monitoring the trends and 552 variability in atmospheric water vapour.

553 Considering five EPN stations, among those with the longest time span, GOPE (Ondrejov, Czech Republic, integrated in the EPN since 31-12-1995), METS (Kirkkonummi, Finland, integrated in 554 the EPN since 31-12-1995), ONSA (Onsala, Sweden, integrated in the EPN since 31-12-1995), 555 PENC (Penc, Hungary, integrated in the EPN since 03 03 2096) and WTZR (Bad Koetzting, 556 Germany, integrated in the EPN since 31 12 1995), we have computed ZTD trends using EPN 557 Repro2, EPN Repro1 completed with the EUREF operational products, radiosonde and ERA-558 Interim data. All of them are also in the IGS Network, for which IGS Repro1 completed with the 559 IGS operational products are available and extracted from the GOP TropDB. First we have 560 removed annual signal from the original time series and marked all outliers according to 3 sigma 561 criteria. Then for all GPS ZTD data sets we have estimated all well-known and recognized shifts 562

**Comment [RVM32]: For which paramete**

**Comment [g33]:** I think you should investigate some more time on the summar of the most important findings of this paper Do not forget that a lot of readers start by reading the abstract and the conclusions an then make up their mind if they proceed wit the rest. In its current form, this summary is not very attracting. Do not forget to mentio the most important findings of sections 2 ar (e.g. one sentence for every subsection of 2 563 
[revised manuscript text omitted]

**Comment [RVM34]:** Please check if the format and journal abbreaviations are compliant with AMT's recommendations

Bruyninx C, Habrich H, Söhne W, Kenyeres A, Stangl G, Völksen C (2012) Enhancement of the
EUREF Permanent Network Services and Products, Geodesy for Planet Earth, IAG Symposia
Series, 136: 27–35. doi: 10.1007/978-3-642-20338

Bruyninx, C., A. Araszkiewicz, E. Brockmann, A. Kenyeres, R. Pacione, W. Söhne, G. Stangl, K.
Szafranek, and Völksen, C.: EPN Regional Network Associate Analysis Center Technical Report
2015, IGS Technical Report 2015, Editors Yoomin Jean and Rolf Dach, Astronomical Institute,
University of Bern, 2015, pp. 101-110, 2015.

COST-716 Exploitation of Ground-Based GPS for Operational Numerical Weather Prediction and
Climate Applications – Final Report, in: Elgered, G., Plag, H.-P., Van der Marel, H., et al. (Eds.),
EUR 21639, 2005.

Dach, R., Hugentobler, U., Fridez, P., and Meindl, M.: Bernese GPS Software Version 5.0, Journal
of Geophysical Research Atmospheres, 119, doi: 10.1002/2013JD021124, 2014.

Dach, R., J. Böhm, S. Lutz, P. Steigenberger and Beutler, G.: Evaluation of the impact of
atmospheric pressure loading modeling on GNSS data analysis, J Geod doi: 10.1007/s00190-0100417-z, 2010.

Dee, D. P., S. M. Uppala, A. J. Simmons, P. Berrisford, P. Poli, S. Kobayashi, U. Andrae, M. A.
Balmaseda, G. Balsamo, P. Bauer, P. Bechtold, and Beljaars, A. C. M.: The ERA-Interim reanalysis:
Configuration and performance of the data assimilation system, Q. J. Roy. Meteor. Soc., 137(656),
553–597, 2011.

Desai, S. D., W. Bertiger, M. Garcia-Fernandez, B. Haines, N. Harvey, C. Selle, A. Sibthorpe, A.
Sibois, and Weiss, J. P.: JPL's Reanalysis of Historical GPS Data from the Second IGS Reanalysis
Campaign, AGU Fall Meeting, San Francisco, CA, 2014.

Dow, J.M., Neilan, R. E., and Rizos, C.: The International GNSS Service in a changing landscape
of Global Navigation Satellite Systems, Journal of Geodesy 83:191–198, doi: 10.1007/s00190-0080300-3, 2009.

Dousa, J. and G.V. Bennett: Estimation and Evaluation of Hourly Updated Global GPS Zenith
Total Delays over ten Months, GPS Solutions, Online publication date: 12-Oct-2012,
doi:10.1007/s10291-012-0291-7, 2012.

Dousa, J. and Vaclavovic P.: The GOP troposphere product from the 2nd European re-processing
 (1996-2014), 2016 (manuscript prepared for AMT)

Gendt, G. SINEX TRO—Solution (Software/technique) INdependent Exchange Format for
combination of TROpospheric estimates Version 0.01, March 1,
1997:https://igscb.jpl.nasa.gov/igscb/data/format/sinex\_tropo.txt, 1997.

Gyori G, and Douša J.: GOP-TropDB developments for tropospheric product evaluation and
monitoring – design, functionality and initial results, In: IAG Symposia Series, Rizos Ch. and
Willis P. (eds), Springer Vol. 143, pp. 595-602., 2016

Guerova, G., J. Jones, J. Douša, G. Dick, S. de Haan, E. Pottiaux, O. Bock, R. Pacione, G. Elgered,
H. Vedel, and M. Bender: Review of the state-of-the-art and future prospective of GNSS
Meteorology in Europe, accepted for publication in to Special Issue: Advanced Global Navigation
Satellite Systems tropospheric products for monitoring severe weather events and climate
(GNSS4SWEC), (AMT/ACP/ANGEO inter-journal SI), 2016.

Comment [RVM35]: Please use consister journal abbreviations. You can find a list of 1 adviced abbreviations on the journal web p

**Comment [RVM36]:** Same remark as her above.

Comment [RVM37]: Submitted to AMT r

- IERS Conventions (2010). Gérard Petit and Brian Luzum (eds.). (IERS Technical Note ; 36)
   Frankfurt am Main: Verlag des Bundesamts für Kartographie und Geodäsie, 2010. 179 pp., ISBN 3 89888-989-6, 2010.
- Ihde, J., Habrich, H., Sacher, M., Söhne, W., Altamimi, Z., Brockmann, E., Bruyninx, C., Caporali,
  C., Dousa, J., Fernandes, R., Hornik, H., Kenyeres, A., Lidberg, M., Mäkinen, J., Poutanen, M.,
  Stangl, G., Torres, J.A., Völksen, C., (2013). EUREF's contribution to national, European and
  global geodetic infrastructures. IAG Symposia, vol. 139, pp. 189–196. doi: 10.1007/978-3-64237222-3\_24.
- Jin, S.G., J. Park, J. Cho, and P. Park: Seasonal variability of GPS-derived Zenith Tropospheric
  Delay (1994-2006) and climate implications, J. Geophys. Res., 112, D09110, doi:
  10.1029/2006JD007772, 2007.
- Haase, J., Calais, E., Talaya, J., Rius, A., Vespe, F., Santangelo, R., Huang, X.-Y., Davila, J. M., Ge,
  M., Cucurull, L., Flores, A., Sciarretta, C., Pacione, R., Boccolari, M., Pugnaghi, S., Vedel, H.,
  Mogensen, K., Yang, X., and Garate, J.: The contributions of the MAGIC project to the COST 716
  objectives of assessing the operational potential of ground-based GPS meteorology on an
  international scale, Physics and Chemistry of the Earth, Part A, 26, 433–437, 2001.
- Haase, J.S., H. Vedel, M. Ge, and E. Calais: GPS zenith troposphteric delay (ZTD) variability in the
  Mediterranean, Phys Chem Earth (A) 26(6–8):439–443, 2001.
- Haase, J., M. Ge, H. Vedel, and Calais, E.: Accuracy and variability of GPS Tropospheric Delay
  Measurements of Water Vapor in the Western Mediterranean, Journal of Applied Meteorology, 42,
  1547-1568, 2003.
- King, R., Herring, T., and Mccluscy, S.: Documentation for the GAMIT GPS analysis software
  10.4., Tech. rep., Massachusetts Institute of Technology, 2010.
- Lutz, S., P. Steigenberger, G. Beutler, S. Schaer, R. Dach, and Jaggi, A.: GNSS orbits and ERPs
  from CODE's repro2 solutions, IGS Workshop Pasadena (USA), June 23–27, 2014.
- Nilsson, T. and Elgered, G.: Long-term trends in the atmospheric water vapor content estimated
   from ground-based GPS data. J. Geophys. Res., 113, doi: 10.1029/2008JD010110, 2008.
- Ning, T., R. Haas, G. Elgered, and. Willén U: Multi-technique comparisons of 10 years of wet delay
  estimates on the west coast of Sweden, J Geod 86: 565. doi: 10.1007/s00190-011-0527-2, 2012.
- Ning, T., J. Wickert, Z. Deng, S. Heise, G. Dick, S. Vey, and Schone, T.: Homogenized time series
  of the atmospheric water vapor content obtained from the GNSS reprocessed data, Journal of
  Climate, doi: 10.1175/JCLI-D-15-0158.1, 2016a
- Ning, T., J. Wang, G. Elgered, G. Dick, J. Wickert, M. Bradke, M. Sommer, R. Querel, and Smale,
  D.: The uncertainty of the atmospheric integrated water vapour estimated from GNSS observations
  Atmos. Meas. Tech., 9, 79-92, doi:10.5194/amt-9-79-2016, 2016b.
- Mangiarotti, S., A. Cazenave, L. Soudarin and Crétaux, J. F.: Annual vertical crustal motions
  predicted from surface mass redistribution and observed by space geodesy, Journal of Geophysical
  Research, 106, B3, 4277, 2001.
- Pacione, R., B. Pace, S.de Haan, H. Vedel, R.Lanotte, and Vespe, F.: Combination Methods of
  Tropospheric Time Series, Adv. Space Res., 47(2) 323-335 doi: 10.1016/j.asr.2010.07.021, 2011.

**Comment [RVM38]:** For Nilsson and Elge you are using J. Geophys. Res. as journal abbreviation. Be consistent, also with the adviced list of journal abbreviations.

- Petrov, L. and Boy, J.-P.: Study of the atmospheric pressure loading signal in very long baseline
  interferometry observations," J. Geophys. Res., 109, B03405, 14 pp., doi: 10.1029/2003JB002500,
  2004.
- Ray, R. D. and Ponte, R. M.: Barometric tides from ECMWF operational analyses, Ann. Geophys.,
  21(8), pp. 1897-1910, doi: 10.5194/angeo-21-1897-2003.
- Saastamoinen, J.: Contributions to the theory of atmospheric refraction, Bull. Geodes., 107, 13–34,
  doi:10.1007/BF02521844, 1973.
- Santerre R.: Impact of GPS Satellite sky distribution. Manuscr. Geod., 16, 28-53, 1991.
- Schmid R, Dach R, Collilieux X, Jäggi A, Schmitz M, Dilssner F (2015) Absolute IGS antenna
  phase center model igs08.atx: status and potential improvements. J Geod 90(4):343–364
- Sohn, D.-H., and Cho, J.: Trend Analysis of GPS Precipitable Water Vapor Above South Korea
  Over the Last 10 Years, J. Astron. Space Sci. 27(3), 231-238 (2010), doi: 10.5140/JASS.2010.27.3.231, 2010.
- Suparta, W.: Validation of GPS PWV over UKM Bangi Malaysia for climate studies, Procedia
   Engineering 50, 325 332, 2012.
- Steigenberger, P., V. Tesmer, M. Krugel, D. Thaller, R. Schmid, S. Vey, and Rothacher, M.:
  Comparisons of homogeneously reprocessed GPS and VLBI long time-series of troposphere zenith
  delays and gradients, J. Geod., 81(6-8), 503–514, doi: 10.1007/s00190-006-0124-y, 2007.
- Tesmer, V., J. Boehm, R. Heinkelmann and Schuh, H.: Effect of different tropospheric mapping
  functions on the TRF, CRF and position time-series estimated from VLBI, Journal of Geodesy June
  2007, Volume 81, Issue 6, pp 409-421, 2007.
- Tregoning, P. and Van Dam, T.: Atmospheric pressure loading corrections applied to GPS data atthe observation level, Geophysical Research Letters, 32, 22, 2005.
- Tregoning P., Watson C.: Atmospheric effects and spurious signals in GPS analyses. J. Geophys.
  Res., 114, B09403, doi: 10.1029/2009JB006344, 2009.
- Van Dam, T., G. Blewitt, and Heflin, M. B.: Atmospheric pressure loading effects on Global
  Positioning System coordinate determinations, Journal of Geophysical Research, 99, B12, 23939,
  1994.
- Vey, S., R. Dietrich, M. Fritsche, A. Rulke, P. Steigenberger, and Rothacher, M.: On the
  homogeneity and interpretation of precipitable water time series derived from global GPS
  observations, J. Geophys. Res., 114, D10101, doi: 10.1029/2008JD010415, 2009.
- Voelksen, C.: An update on the EPN Reprocessing Project: Current Achievements and Status,
  Presented at EUREF 2011 Symposium, Chisinau, Republic of Moldova, May 25-28 2011,
  http://www.epncb.oma.be/\_documentation/papers/eurefsymposium2011/an\_update\_on\_epn\_reproc
  essing\_project\_current\_achievement\_and\_status, 2011.
- Wang, J., Zhang, L., Dai. A., Van Hove, T., Van Baelen, J.: A near-global, 2-hourly data set of
  atmospheric precipitable water dataset from ground-based GPS measurements, J Geophys Res
  112(D11107). doi:10.1029/2006JD007529, 2007.

Wang, J. and Zhang, L.: Climate applications of a global, 2-hourly atmospheric precipitable water
dataset derived from IGS tropospheric products, J Geod 83: 209. doi: 10.1007/s00190-008-0238-5,
2009.

789 Webb, F. H., and Zumberge, J.F.: An Introduction to GIPSY/OASIS II. JPL D-11088, 1997.

Vedel, H., K. S. Mogensen, and X.-Y. Huang: Calculation of zenith delays from meteorological
data comparison of NWP model, radiosonde and GPS delays, Phys. Chem. Earth Pt. A, 26, 497–
502, doi: 10.1016/S1464-1895(01)00091-6, 2001.

Zus, F, Dick, G, Heise, S, Dousa, J, and Wickert J.: The rapid and precise computation of GPS slant
total delays and mapping factors utilizing a numerical weather model, Radio Sci, 49(3): 207-216,
doi: 10.1002/2013RS005280, 2014.

**797 Table**

**798 Table Captions**

| 799 | Table 1: EPN Analysis Centres providing EPN-Repro2 solutions Table 1: EPN Analysis Centres           |            | Formatted  |
|-----|------------------------------------------------------------------------------------------------------|------------|------------|
| 800 | providing EPN Repro2EPN Repro2 solutions.                                                            |            | Formatted  |
| 801 | Table 2: EPN-Repro2 processing options for each contributing solutions. AS0 solution is provided     |            | Formatted  |
| 802 | by ASI/CGS (Matera, Italy), GO0, GO1 and GO4 solutions are provided by GOP (Pecny, Czech             | $\searrow$ | Formatted  |
| 803 | Republic), IGO solution by IGE (Madrid, Spain), LPO and LP1 solutions by LPT (Waben,                 |            | Formatted  |
| 804 | Switzerland), and MU2 and MU4 solutions by MUT (Warsaw, Poland). Table 2: EPN Repro2EPN-             |            | and gramm  |
| 805 | Repro2 processing options for each contributing solutions. AS0 solutions provided by ASI/CGS         |            |            |
| 806 | (Matera, Italy), GO0, GO1 and GO4 solutions provided by GOP (Pecny, Czech Republic), IG0             |            |            |
| 807 | solution provided by IGE (Madrid, Spain), LPO and LP1 solutions provided by LPT (Waben,              |            |            |
| 808 | Switzerland), MU2 and MU4 solutions provided by MUT (Warsaw, Poland).                                |            |            |
| 809 | Table 3. Percentage of red, orange and yellow biases (see text) for each contributing solution Table |            | Formatted  |
| 810 | 3. Percentage of red, orange and yellow bias for each contributing solution.                         |            | and gramm  |
| 011 | Table 4. Mean statistics and uncertainties, calculated from results of individual stations, provided | _          | - 1 |
| 011 | Table +: Wear statistics and uncertainties, carculated non-results of individual stations, provided  |            | Formatted  |
| 812 | for AC individuals and EUREF combined (Repro1 and Repro2) tropospheric parameters compared           |            |            |
| 813 | to the ERA-Interim re-analysis (EGRD = east gradient, NGRD = north gradient) Table 4. Mean           |            |            |
| 814 | statistics and uncertainties, calculated from results of individual stations, provided for AC        |            |            |
| 815 | individuals and EUREF combined (Repro1 and Repro2) tropospheric parameters compared to the           |            |            |
| 816 | ERA Interim re-analysis.                                                                             |            |            |
|     |                                                                                                      |            |            |

| Formatted: English (U.K.)                                 |
|-----------------------------------------------------------|
| Formatted: Font: Not Italic, English (U.K.                |
| Formatted: English (U.K.)                                 |
| Formatted: English (U.K.)                                 |
| Formatted: Font: Not Italic, Check spellin
and grammar |

[revised manuscript text omitted]

Figure 8, VENE (Venice Italy) time series of total consistency (for definition, see Fig. 7) in the up component for the period July 21st, 1996 - July 28, 2007 (GPS weeks 0863-1437). Figure 8 VENE

| Formatted: Font color: Auto                                                         |  |  |  |  |  |
|-------------------------------------------------------------------------------------|--|--|--|--|--|
| Formatted: Font: Not Italic, Font color:
Auto                                    |  |  |  |  |  |
| Formatted: Font color: Auto, Do not cho
spelling or grammar                      |  |  |  |  |  |
|                                                                                     |  |  |  |  |  |
| Formatted: Font color: Auto                                                         |  |  |  |  |  |
| Formatted: Font: Not Italic, Font color:
Auto, Check spelling and grammar |  |  |  |  |  |
| Formatted: Font color: Auto                                                         |  |  |  |  |  |
| Formatted: Font: Not Italic                                                         |  |  |  |  |  |
| Formatted: Font: Font color: Black                                                  |  |  |  |  |  |
|                                                                                     |  |  |  |  |  |
| Formatted: Not Superscript/ Subscript                                               |  |  |  |  |  |
| Formatted: Not Superscript/ Subscript                                               |  |  |  |  |  |

| Formatted: Font color: Auto                                                         |
|-------------------------------------------------------------------------------------|
| Formatted: Font: Not Italic, Font color:
Auto, Check spelling and grammar |
| Formatted: Font color: Auto                                                         |
| Formatted: Font color: Auto, English (U.                                            |
| Formatted: Font color: Auto                                                         |
| Formatted: Font: Italic                                                             |
| Formatted: Font: Not Italic                                                         |
| Formatted: Not Superscript/ Subscript                                               |
|                                                                                     |
| Formatted: Font: Not Italic, Check spelli and grammar                               |
| Formatted: English (U.K.)                                                           |
| Formatted: Font: Not Italic, English (U.K
Check spelling and grammar      |
| Formatted: English (U.K.)                                                           |
| Formatted: English (U.K.)                                                           |
| Formatted: Font: Not Italic, English (U.K
Check spelling and grammar             |
| Formatted: English (U.K.)                                                           |
| Formatted: English (U.K.)                                                           |
| Formatted: Not Superscript/ Subscript                                               |
|                                                                                     |

| 874
875                      | (Venice Italy) time series of total consistency in up component for the period July 21st, 1996 July 28, 2007 (GPS week 0863-1437).                                                                                                                                                                                                                                                                                                                                       |   |                                                                                                                         |
|---------------------------------|--------------------------------------------------------------------------------------------------------------------------------------------------------------------------------------------------------------------------------------------------------------------------------------------------------------------------------------------------------------------------------------------------------------------------------------------------------------------------|---|-------------------------------------------------------------------------------------------------------------------------|
| 876
877                      | Figure 9 EPN station CAGL (Cagliari, Sardinia Island, Italy). Upper part: Radiosondes (in red) and GPS (in blue) ZTD time series. Lower part: ZTD differences, calculated as RS minus GPS. Figure 9                                                                                                                                                                                                                                                                      |   | Formatted: Font: Not Italic, Check spellin and grammar                                                                  |
| 878
879                      | EPN station CAGL (Cagliari, Sardinia Island, Italy). Upper part: Radiosondes (in red) and GPS (in blue) ZTD time series. Lower part differences.                                                                                                                                                                                                                                                                                                                         |   | Formatted: English (U.K.)                                                                                               |
| 880
881
882
883        | Figure 10: RS minus GPS ZTD biases for all GPS-RS station pairs. The error bar is the standard deviation. Sites are sorted with increasing distances from the nearest radiosonde launch site Figure 10 GPS versus Radiosonde Bias. The error bar is the standard deviation. Sites are sorted according to the increasing distances from the nearest Radiosonde launch site.                                                                                              |   | Formatted: Font: Not Italic, English (U.K.
| 884
885
886
887        | Figure 11: Distributions of station mean ZTD biases (left) and standard deviations (right) of EPN-
Repro1 and Repro2 compared to ERA-Interim. Figure 11: Distributions of station means (left) and
standard deviations (right) of EPN Repro1EPN Repro1 and Repro2 ZTDs compared to ERA-
Interim ZTDs.                                                                                                                                                           |   | Formatted: Font: Not Italic, Check spelli
and grammar                                                                |
| 888
889
890               | Figure 12 : Site-by-site ZTD improvements of EPN-Repro2 versus EPN-Repro1 compared to ERA-
InterimFigure 12: Site by site ZTD improvements of EPN Repro2 EPN Repro2 versus EPN
Repro1 EPN Repro1 compared to ERA Interim                                                                                                                                                                                                                      | _ | Formatted: Font: Not Italic, Check spelli
and grammar                                                                |
| 891
892
893
894
895 | Figure 13: Time series of monthly mean biases (lower part) and standard deviations (upper part) for ZTD differences between EPN-Repro2 and ERA-interim re-analysis (GPS minus ERA-interim?). Uncertainties are calculated over all stations. Figure 13: Time series of monthly mean biases (lower part) and standard deviations (upper part) for ZTD differences of EPN-Repro2EPN-Repro2 and NWMERA interim re-analysis. Uncertainties are calculated over all stations. |   | Formatted: Font: Not Italic, Check spellin
and grammar                                                               |
| 896
897
898               | Figure 14 : Geographical distribution of ZTD biases (left) and standard deviations (right) for EPN-
Repro2 compared to ERA-Interim . Figure 14: Geographical display of ZTD biases (left) and standard deviations (right) for EPN Repro2 EPN Repro2 products compared to the ERA Interim.                                                                                                                                                        |   | Formatted: Font: Not Italic, Check spellin
and grammar                                                               |
| 899
900
901               | Figure 15: ZTD trend comparisons at five EPN stations for 5 different ZTD datasets. The error bars are the formal errors of the estimated trend values. Figure 15: ZTD trend comparisons at five EPN stations. The error bars are the formal error of the trend values.                                                                                                                                                                                                  |   | Formatted: Font: Not Italic, Check spelli
and grammar                                                                |
| 902                             |                                                                                                                                                                                                                                                                                                                                                                                                                                                                          |   |                                                                                                                         |

Figure 1. Time series of the number of GNSS observations for the period 1996-2014. GPS
 observations are shown in red, GPS+GLONASS in blue and their differences in green. The
 difference becomesis significant starting from 2008.

---

## Author Response (AR2)

**Authors' Response**

The authors would like to thank the Associate Editor for his further revision.

His comments and suggestions have improved the quality of the manuscript.

In the revised version, we have taken all of them into account and we have replied to his comments.

EPN-Repro2: A reference GNSS tropospheric dataset over Europe.

[revised manuscript text omitted]

**Commentato [g22]:** This is the radiosonde type: Vaisala RS90.

**Commentato [g23]:** M2K2 is a radiosonde type that Olivier Bock has assessed in his AMT paper (doi:10.5194/amt-6-2777-2013). Please comment if in this paper also a positive bias has been found.

**Commentato [AA24]:** Bock et al. (2013) report the dry bias for M2K2 radiosonde type. With respect to their results, our positive bias should be considered as moist bias and is not consistent with them. However, Bock et al. also noted that their results are not consistent with the nearby station in France.

was found at a French site. However, they also indicated that their results are not consistent with another nearby radiosonde station and needs further investigation. Further investigation in our study is also needed for several near or moved GPS stations, or switched radiosonde type at one station. For example in Brussels (Belgium) BRUS station, included in the EPN network since 1996, was replaced by BRUX in 2012. Their bias w.r.t. radiosonde (WMO code 6447) has opposite sign (-1.2 mm and 3.4 mm respectively). However, the radiosonde type was switched from RS80 to RS90 in 2007 (Van Malderen et al., 2014), which makes the bias for BRUS additionally affected by the change of the radiosonde type. .

In agreement with Ning et al. (2012), the ZTD standard deviation generally increases with the distance from the radiosonde launch site. It is in the range of [0.16; 0.76] %, which corresponds to [3; 18] mm in ZTD, till 15 km (first band in Figure 10); in [0.29; 0.78] %, corresponding to [7; 19] mm, till 70 km (second band in Figure 10), and in [10; 33] mm till 133 km (third band in Figure 10). The numbers of the standard deviation are comparable with previous studies. Haase et al. (2001) showed a very good agreement with biases less than 5 mm in ZTD and a standard deviation of 12 mm for most of the analysed sites in Mediterranean. Similar results (6.0 mm ± 11.7 mm) were obtained also by Vedel et al. (2001). Both studies were based on non-collocated pairs at sites less than 50 km from each other. Pacione et al (2011), considering 1-year of GPS ZTD and radiosonde data over the E-GVAP super sites network, obtained a standard deviation of 5-14 mm. Dousa et al. 2012 evaluated ZTDs from GNSS and radiosondes on a global scale over a 10-month period and reported a standard deviation of 5–16 mm.

If we compare both the EPN-Repro1 ZTD product (completed with the EUREF operational product after 30 December 2006) and the EPN-Repro2 with the radiosonde ZTDs for the same period 1996-2014, we found an improvement of approximately 3-4% in the overall standard deviation for the second processing.

**4.2    Evaluation versus ERA-Interim data**

We also compared the EPN-Repro2 ZTDs with the ZTDs calculated from ERA-Interim (Dee et al., 2011) from the European Centre for Medium-Range Weather Forecasts (ECMWF). The ERA-Interim is a re-analysis product of a Numerical Weather Prediction (NWP) model and is available every 6 hours (00, 06, 12, 18 UTC) with a horizontal resolution of $1 \times 1$ degree and with 60 vertical model levels.

For the period 1996-2014 and for each EPN station, the ZTD and tropospheric linear horizontal gradients were computed using the GFZ (German Research Centre for Geosciences) ray-tracing software (Zus et al., 2014). Combined EUREF Repro1 and Repro2 products as well as individual

**Commentato [g25]:** VRS80L is the radiosonde type: Vaisala RS80.

**Commentato [g26]:** In our AMT2014 paper, section 4.2, we also make the RS-GPS comparison separately for the two (or 3) radiosonde types that have been used at Brussels. As a matter of fact, we switched from RS80 to RS90 in August 2007. You are hence comparing BRUS with RS80 and RS90/RS92, while comparing BRUX only with RS92. This will have a non-neglible effect on the comparison, and is hence not related only to the change of the GPS station!

**Commentato [AA27R26]:** Included in the text.

[revised manuscript text omitted]

**Commentato [g54]:** In this figure, you use ZPD instead of ZTD. You do not explain what it stands for and this is also very inconsistent with the rest of the paper. Please change the titles and the label in the figure!

**Commentato [PR55R54]:** Done

[Figure]

Figure 10: GPS minus RS ZTD biases for all GPS-RS station pairs. The error bar is the standard deviation. Sites are sorted with increasing distances from the nearest radiosonde launch site. The x-axis shows the GPS station and the radiosonde site WMO code.

[Figure]

Figure 11: Distributions of station mean ZTD biases (left) and standard deviations (right) of EPN-
Repro1 and Repro2 compared to ERA-Interim.

[Figure]

Figure 12: Site-by-site ZTD improvements of EPN-Repro2 versus EPN-Repro1 compared to ERA-
Interim.

[Figure]

Figure 13: Time series of monthly mean biases (lower part) and standard deviations (upper part) for
ZTD differences between EPN-Repro2 and ERA-Interim re-analysis (GPS minus ERA-interim).
Uncertainties are calculated over all stations.

Commentato [RVM56]: Please confirm or modify.

Commentato [PR57R56]: yes

[Figure]

Figure 14: Geographical distribution of ZTD biases (left) and standard deviations (right) for EPN-
Repro2 compared to ERA-Interim.

[Figure]

Figure 15: ZTD trend comparisons at five EPN stations for 5 different ZTD datasets. The error bars are the formal errors of the estimated trend values.

---

## Author Response (AR3)

**Authors' Response**

The authors would like to thank the Associate Editor for his further revision.

His comments and suggestions have improved have been taken into account in the reviewed manuscript.

As far as the biases are concerned with respect to other techniques, they are computed as RS-GNSS and ERA-GNSS. Text, tables and figures are reviewed accordingly.